# Learning-to-Context Slope:
# Evaluating In-Context Learning Effectiveness Beyond Performance Illusions

## Abstract

In-context learning (ICL) has emerged as an effective approach to enhance the performance of large language models (LLMs). However, its effectiveness varies significantly across models and tasks, posing challenges for practitioners to determine when ICL reliably improves performance. Current evaluation approaches, reliant on performance change after applying ICL, suffer from low reliability, poor attribution, and impracticality in data-insufficient scenarios. We propose the **Learning-to-Context Slope (LCS)**, a novel metric that quantifies ICL effectiveness by modeling the slope between *learning gain* (loss decrease from demonstrations) and *contextual relevance* (demonstration-input relevance). LCS addresses key limitations of performance-based metrics: *(i)* it captures continuous loss changes even when outputs are incorrect, improving reliability; *(ii)* its formulation attributes ICL failures to weak contextual alignment (inability to adapt inputs to demonstrations) or strong output calibration (self-verification of correctness); and *(iii)* it minimizes reliance on labeled data via synthetic evaluation. Extensive experiments demonstrate that LCS strongly correlates with performance improvements in labeled settings and reliably reflects true effectiveness in biased or data-scarce scenarios. Further analysis reveals actionable thresholds for LCS and identifies model capabilities critical to ICL success[1].

## 1 Introduction

In-context learning (ICL) has emerged as a popular and effective paradigm for enhancing large language model (LLM) performance across diverse tasks, as it eliminates the need to retrain the LLMs (Brown et al., 2020; Dong et al., 2024). By incorporating task-specific demonstrations into the input, ICL enables LLMs to adapt to specific tasks and generate more accurate outputs without parameter updates. Recently, several efforts have been made on unveiling the underlying mechanisms of ICL (Zhou et al., 2024; Edelman et al., 2024a; Park et al., 2025) and exploring methods to further boost the ICL performance (Wang et al., 2023b; Rubin et al., 2022; Agarwal et al., 2024).

However, as illustrated in Figure 1a, even on the models with strong ICL capability like Llama3.1 (Grattafiori et al., 2024), ICL fails to enhance, and in some cases even harms, the performance (DeepSeek-AI et al., 2025; Huang & Wang, 2025; Zheng et al., 2025), showing different ICL effectiveness across different models. This observation raises a critical question: **How can practitioners reliably determine whether ICL is effective for a given model on a specific task**? This uncertainty poses practical challenges in the real-world deployment of ICL:

- For tasks *with labeled data*, practitioners often attempt to evaluate ICL effectiveness by observing performance changes after applying the selected demonstrations. However, this approach suffers from two critical limitations. *(i) Low Reliability*: Performance fluctuations may stem from various factors like the quality of the instruction and selected demonstrations, making it difficult to isolate whether ICL itself is ineffective. *(ii) Poor Attribution*: Disentangling the impact of individual factors requires costly repeated evaluations, hindering actionable analysis and insights.
- For tasks *without labeled data*, there is no direct way to assess whether adding demonstrations for ICL actually improves outcomes, leaving practitioners without clues for improvements.

---

[1]Our code and data will be released upon acceptance.

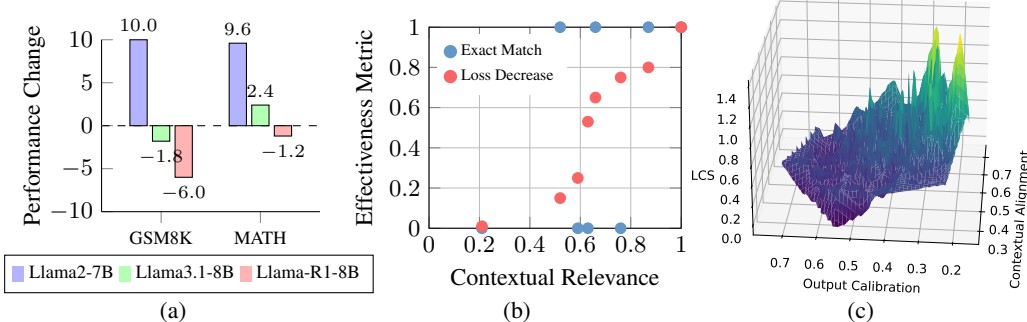

Figure 1: (a) Performance change of different models before and after applying ICL, where ICL exhibits varying effectiveness across different models on the same dataset. (b) Comparisons between metrics based on exact match and loss decrease. Each dot denotes an example data of MATH using Llama3.1-8b with different demonstrations. Performance-based metrics with only binary values fail to quantify the varying contributions of different demonstrations to achieving correct results. In contrast, metrics based on loss decrease yield continuous values, enabling better reliability on measuring ICL effectiveness. (c) The impact of the contextual alignment and output calibration capabilities of the model on the LCS metric.

In light of these challenges, we propose a novel metric, named **L**earning-to-**C**ontext **S**lope (**LCS**), which quantifies the ICL effectiveness by capturing the *slope between the loss decrease by demonstrations (**learning gain**) and the demonstration relevance to the user input (**contextual relevance**)*. Specifically, LCS is grounded in the perspective of the loss decrease in ICL (Wang et al., 2024b; Yang et al., 2024). For a given model and task, it evaluates how the learning gain varies with demonstrations of different contextual relevance. This metric explicitly captures the two most important elements in *in-context learning*: *learning* and *context* (Dong et al., 2024). When ICL effectiveness is high, even demonstrations with low relevance can yield a significant loss decrease. Conversely, when ICL effectiveness is low, the change of learning gain with demonstration relevance is marginal.

Compared to performance-based measurement, LCS offers the following advantages: *(i) Higher Reliability*: As shown in Figure 1b, even when ICL fails to produce correct answers for user inputs, LCS can still capture continuous changes in model loss, providing a more reliable reflection of ICL effectiveness. *(ii) Better Attribution*: LCS is grounded in an intuitive mathematical formulation, enabling clearer analysis of how different factors influence ICL effectiveness. As shown in Figure 1c, ICL tends to be ineffective when 1) the model fails to recognize the relevance of the demonstration to the input (*i.e.*, the *contextual alignment capability*), or 2) the model can independently verify the correctness of the output to the user input without adding demonstrations (*i.e.*, the *output calibration capability*). *(iii) Reduced Reliance on Labeled Evaluation Data*: We theoretically show that LCS derived from synthetic data is consistently lower than that obtained from real data, and empirically identify a threshold value of LCS indicative of effective ICL. Even in data-insufficient scenarios, LCS can still offer actionable insights into ICL effectiveness.

Our contributions can be summarized as follows:

- We propose a novel metric, namely Learning-to-Context Slope (LCS), to measure the ICL effectiveness by capturing the two most important elements in ICL, including the learning gain and the contextual relevance of the demonstrations.

- To validate the effectiveness of LCS, we conduct extensive experiments on eight mainstream datasets covering mathematics, code, reasoning, and domain-specific tasks (*e.g.*, finance and e-commerce). The results validate a strong positive correlation between LCS and task performance improvements in scenarios where abundant labeled data enables reliable performance-based evaluation. When labeled data exhibits inherent biases that distort performance-based metrics, LCS consistently reflects true ICL effectiveness, underscoring its reliability. Even without labeled data, LCS provides actionable insights into ICL effectiveness by leveraging synthetic data.

- Further analysis identifies two key factors in LLMs that hinder ICL effectiveness: 1) weak contextual alignment capability to adapt inputs to task-specific demonstrations, and 2) strong output calibration capability to independently verify the correctness of outputs.

## 2 PROPOSED METRIC: LEARN-TO-CONTEXT SLOPE

We introduce a novel metric, named Learn-to-Context Slope (LCS), to measure the ICL effectiveness. First, we interpret the ICL effectiveness by measuring the loss decrease brought by using demonstrations based on the Bayesian model (§2.1). Then, we present our LCS metric to measure the ICL effectiveness, based on which we discuss two main factors that influence the ICL effectiveness (§2.2). Further, we discuss the relationship between the metric using synthetic data and real data, aiding the application under the data-insufficient scenario (§2.3). We discuss why the analysis is based on conditional probability in Appendix E.4.

### 2.1 INTERPRETING ICL EFFECTIVENESS VIA LOSS DECREASE

Motivated by previous studies (Wang et al., 2024b; Yang et al., 2024), the ICL effectiveness of a given predictive distribution $p$ with the parameter $\theta$ on a specific task with the task $\mathcal{C} = (Q, X, D)$ can be measured by the generation loss, *i.e.,* negative log-likelihood:

$$\mathbb{L}_\theta(X|Q; D) = -\log p(X|Q; D), \tag{1}$$

where $Q$ denotes the user input, $X$ represents the labeled output corresponding to $Q$, and $D$ denotes the demonstration, which are the random variables in the sampling spaces $\mathcal{X}$, $\mathcal{Y}$, and $\mathcal{D}$, respectively. Based on the Bayesian model (Zhang et al., 2025; Jesson et al., 2025), this loss can be obtained by:

$$\mathbb{L}_\theta(X|Q; D) = \mathbb{L}_\theta(X|Q) - (\log p(D|Q; X) - \log p(D|Q)), \tag{2}$$

where $\mathbb{L}_\theta(X|Q)$ represents the loss of zero-shot generation, which is fixed given the model and task. The proof of Equation 2 is presented in Appendix C.1. It can be observed that only the second term, *i.e.,* $\log p(D|Q; X) - \log p(D|Q)$, is relevant to the demonstrations, specifically the reduction in loss brought about by the demonstrations. Intuitively, this term also measures the information of the user output $X$ that helps to decide the demonstration $D$.

### 2.2 METRIC OF ICL EFFECTIVENESS: LCS

For simplicity, we denote the **Learning Gain** brought by the demonstrations as $I_p(X \to D|Q) = p(D|Q; X) - p(D|Q)$ to reflect the decrease in loss, we discuss it in detail in Appendix E.5. To evaluate the overall effectiveness of the given specific model and task in ICL, we propose measuring effectiveness by assessing how the learning gain varies with demonstrations of different relevance. The motivation is that even demonstrations with low relevance to the user question can still lead to significant learning gains for tasks and models where ICL is highly effective. Conversely, when the ICL effectiveness is low, the change in learning gain with demonstration relevance is marginal. We measure the **Contextual Relevance** of the demonstration to the user question as $I_p(D \to X|Q) = p(X|Q; D) - p(X|Q)$. The contextual relevance is quantified by how much information for inferring the output $X$ can be learned from the demonstration $D$ in the context. To demonstrate that the contextual relevance defined by probability can reflect the demonstration relevance, we also compare it with other relevance measurement of the demonstration to the user question in Appendix F.2.

We have that the learning gain $I_p(X \to D|Q)$ and the contextual relevance $I_p(D \to X|Q)$ satisfy:

**Theorem 1.**

$$I_p(X \to D|Q) = \frac{p(D|Q)}{p(X|Q)} I_p(D \to X|Q)$$

The proof of Theorem 1 is presented in Appendix C.2. According to the theorem, learning gain and contextual relevance are positively correlated with a certain slope. A larger slope indicates a greater decrease in loss when increasing the information relevant to the user question of the demonstrations, thereby making ICL more effective.

In practice, let $\hat{p}$ represent the empirical probability distribution and $C = \{(q_i, x_i, d_i)\}_n$ be the sampling on $\mathcal{C}$, we calculate the slope of Theorem 1 ($r_{\hat{p}}$) on $C$ with the least squares method (Wang

et al., 2018):

$$r_{\hat{p}} = \frac{\sum_{i=1}^{n}(t_i - \bar{t})(s_i - \bar{s})}{\sum_{i=1}^{n}(t_i - \bar{t})^2}, \texttt{where}$$

$$s_i = I_{\hat{p}}(x_i \to d_i | q_i), t_i = I_{\hat{p}}(d_i \to x_i | q_i) \tag{3}$$

$$\bar{s} = \frac{1}{n}\sum_{i=1}^{n} s_i, \bar{t} = \frac{1}{n}\sum_{i=1}^{n} t_i.$$

We use $r_{\hat{p}}$ as the metric to measure the ICL effectiveness, which we call the **Learning-to-Context Slope (LCS)**. Although $r_p = \frac{p(D|Q)}{p(X|Q)}$, considering that $\hat{p}$ has the error compared with $p$, $r_{\hat{p}} \neq \frac{\hat{p}(D|Q)}{\hat{p}(X|Q)}$. In Appendix C.3, we discuss the impact of error and prove that the impact of the error on $r_{\hat{p}}$ is less than $\frac{\hat{p}(D|Q)}{\hat{p}(X|Q)}$. We discuss how to calculate our metric in detail in Appendix D.2.

Based on Theorem 1, it can be observed that there are two main factors influencing the ICL effectiveness: *the contextual alignment capability* that learn the question-relevant information from the demonstrations ($\hat{p}(D|Q)$), and *the output calibration capability* that verify the correctness of the output to the given input ($\hat{p}(X|Q)$). Therefore, given a specific model and task, the reasons for poor ICL effectiveness can be attributed to two aspects: *(i) Low Contextual Alignment Ability*: The model fails to adequately comprehend the task-relevant information in the provided demonstrations. *(ii) High Output Calibration Capability*: The model possesses a strong inherent ability to verify the input consistency with the given output. We further discuss the meaning of the contextual alignment capability and the output calibration ability in detail in Appendix E.1.

### 2.3 ICL Effectiveness without Labeling

Since the calculation of LCS in § 2.2 relies on labeled data, its application to new tasks in data-insufficient scenarios is limited. Prior work has shown that the resource requirements for obtaining task questions are lower than those for obtaining the labels (Shen et al., 2019; Tan et al., 2024). Therefore, in this section, we discuss the relationship of LCS with synthetic data and real data, using only the labeled input, which satisfies the following:

**Theorem 2.** *Let $\hat{D}, D^*$ denote two demonstration satisfying that, for all $X \sim \mathcal{X}$ and $Q \sim \mathcal{Q}$:*

$$\hat{p}(\hat{D} \mid Q; X) \leq \hat{p}(D^* \mid Q; X).$$

*The above condition means that demonstration $D^*$ is better than $\hat{D}$ to help to generate the correct answer. Then, we can derive that:*

$$\frac{\hat{p}(\hat{D}|Q)}{\hat{p}(\hat{X}|Q)} \leq \frac{\hat{p}(D^*|Q)}{\hat{p}(X^*|Q)}$$

The conclusion in Theorem 2 suggests that the more demonstrations that can help the model make correct predictions, the larger the corresponding LCS. Considering that previous work has shown that the quality of synthesized data is generally lower than annotated data (Ashok & May, 2025; Gulati et al., 2023), we can consider $\hat{D}$ as a synthesized demonstration and $D^*$ as an annotated demonstration. Therefore, we can observe that LCS fitted with synthetic data is consistently smaller than that using real data. Consequently, while fitting synthetic data can reflect the ICL effectiveness to some extent, the magnitude of the effectiveness derived is lower than that of the real effectiveness.

## 3 Experiment

In this section, we empirically investigate three research questions about the ICL effectiveness: **RQ1**. How to reliably evaluate the ICL effectiveness? **RQ2**. How do different factors influence the ICL effectiveness? **RQ3**. Can synthetic data accurately reflect the ICL effectiveness?

| Dataset | Llama2-7b | | Llama3.1-8b | | Llama-R1-8b | |
|---|---|---|---|---|---|---|
| | Δ | LCS | Δ | LCS | Δ | LCS |
| GSM8K | +10.0 | 0.32 | −1.8 | 0.07 | −6.0 | 0.05 |
| MATH | +9.6 | 1.03 | +2.4 | 0.34 | −1.2 | 0.09 |
| HumanEval | −0.6 | 0.07 | −2.5 | 0.10 | −3.0 | −0.11 |
| MBPP | −0.5 | 0.05 | +0.8 | 0.15 | −6.4 | 0.07 |
| ARC-C | +11.6 | 0.74 | −1.9 | −0.54 | −0.3 | 0.08 |
| MMLU-Pro | +5.5 | 0.64 | +2.6 | 0.52 | −5.5 | −0.04 |
| FinQA | +7.3 | 0.63 | +4.9 | 0.82 | −1.8 | 0.04 |
| Amazon | +0.5 | 0.07 | +5.0 | 0.94 | +11.8 | 0.37 |

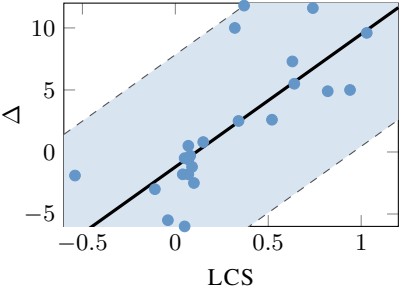

Table 1: Performance and LCS across different models and datasets. Δ denotes the performance change of 1-shot compared to 0-shot. Results in green indicate a significant improvement with ICL, while those in red indicate no improvement or performance drop. Detailed performance is presented in Appendix F.1.

Figure 2: The performance improvement Δ brought by ICL (y-axis) with different LCS (x-axis) on different models and datasets. The solid line in the graph represents the fitted line for all data points. The Pearson correlation coefficient is 0.737.

### 3.1 EXPERIMENT SETUP

**Dataset** We conduct experiments on four mainstream tasks: math (GSM8K (Cobbe et al., 2021), MATH (Hendrycks et al., 2021)), code (HumanEval (Chen et al., 2021a), MBPP (Austin et al., 2021)), reason (ARC-Challenge (Yadav et al., 2019), MMLU-Pro (Wang et al., 2024d)), and domain-specific (FinQA (Chen et al., 2021b), Amazon Review (Ni et al., 2019)). We introduce the above dataset, as well as the split of the demonstrations and the test data, in Appendix D.1.

**Metric** For datasets of math, reasoning, and domain-specific, we use Exact Match (EM) (Cobbe et al., 2021) as the evaluation metric. For the datasets of code, we use Pass@1 (Chen et al., 2021a) as the metric. To prove that LCS is a better metric than performance change to reflect the ICL effectiveness, our main experiment includes two parts: *(i)* In §3.2.1, we evaluate that when performance change can reflect the effectiveness of ICL, LCS can also reflect the effectiveness of ICL. *(ii)* In §3.2.2, we present that LCS can still reflect the ICL effectiveness, even when the provided demonstrations do not lead to performance improvements.

**Model** We conduct our experiments on three mainstream LLMs: Llama2-7b (Touvron et al., 2023), Llama3.1-8b (Grattafiori et al., 2024), and DeepSeek-R1-Distill-Llama-8b (Llama-R1-8b) (DeepSeek-AI et al., 2025), which cover different ICL capabilities to fully evaluate whether our metric can reflect the ICL effectiveness. We also conduct experiments on models of other scales and series in Appendix F.3, further validating the effectiveness of LCS. We discuss how to adapt LCS to black-box LLMs in Appendix F.6.

**Implementation Details** We evaluate the performance on all datasets under 0-shot and 1-shot settings, using BM25 to select the demonstrations for each user input. We also discuss the performance and ICL effectiveness under different shots in §3.3.3. Following DeepSeek-AI et al. (2025), we set the maximum generation length to 32,768. Our experiments are conducted on a single A100-80G, with an average computation time of approximately 20 minutes on each dataset and model.

### 3.2 RQ1. HOW TO RELIABLY EVALUATE THE EFFECTIVENESS OF ICL?

First, we discuss that LCS can accurately reflect the effectiveness of ICL. Subsequently, we provide experimental evidence demonstrating that performance improvement is insufficient for accurately reflecting the ICL effectiveness. In addition, we present that LCS can reflect the performance improvement brought by ICL to a certain extent.

### 3.2.1 LCS RELIABLY REFLECTS THE ICL EFFECTIVENESS

According to the main experimental results shown in Table 1, there are several notable observations:

**The ICL effectiveness is independent of dataset difficulty.** For Llama2-7b, ICL is effective on the MATH dataset but fails on the easier Amazon Review dataset. Conversely, for Llama-R1-

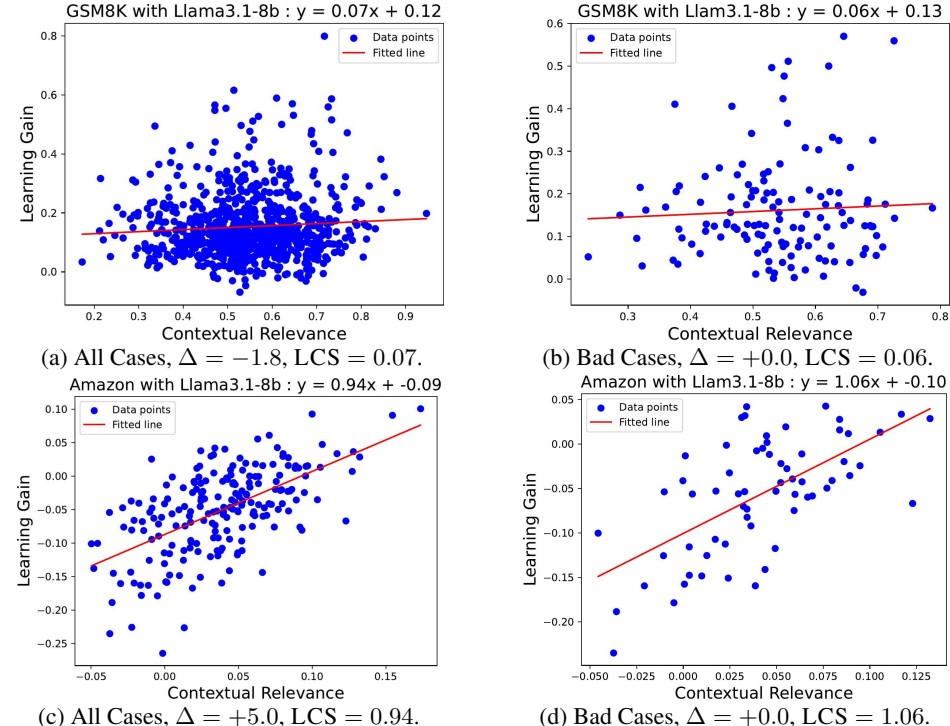

Figure 3: The experimental results of using Llama3.1-8b on GSM8K and Amazon under the full set and the bad cases of ICL. $\Delta$ denotes the performance change of ICL compared with zero-shot.

8b, ICL is ineffective on MATH but performs well on Amazon Review. This discrepancy arises because, for more difficult datasets, the model struggles to comprehend the relationships between demonstrations, answers, and user questions, leading to a decline in both the ICL ability and the answer verification ability. Consequently, it is uncertain whether LCS increases or decreases on more difficult datasets, supporting that ICL effectiveness is irrelevant to the difficulty of the dataset.

**The ICL effectiveness is independent of model capability.** In Amazon Review, Llama-R1-8b demonstrates a significant improvement with ICL, whereas the less capable Llama2-7b does not exhibit a noticeable performance improvement. This discrepancy arises because, as the model capability increases, both the contextual alignment capability and the output calibration capability increase simultaneously, making it uncertain whether LCS rises or falls.

**The performance improvement brought by ICL is positively related to LCS.** To evaluate whether LCS effectively reflects the efficacy of ICL, we analyze the performance improvement with different LCS, as illustrated in Figure 2. A high LCS suggests that the model achieves higher learning gain as the contextual relevance increases, demonstrating that LLMs learn how to solve the task from the demonstrations, thereby improving performance. In contrast, a low LCS indicates that the learning gain from the demonstrations remains relatively constant regardless of the contextual relevance, implying limited learning from the demonstrations. Notably, the relationship between the change in EM and LCS is not strictly linear. Since the factors influencing EM are complex and difficult to formalize, in this paper, we only conclude that LCS is positively correlated with the change in EM. We discuss the empirical threshold of the ICL effectiveness using LCS in Appendix E.2.

### 3.2.2 THE PERFORMANCE CHANGE CANNOT REFLECT THE ICL EFFECTIVENESS

In §3.1, we assume that whether performance improves or not can genuinely reflect the ICL effectiveness. However, in practical applications, the quality of demonstrations or instructions can impact performance, causing no performance improvement even for models and tasks where ICL is effective. To demonstrate that LCS can still reflect the ICL effectiveness even when performance does not improve, we plot $r_{\hat{p}}$ on the bad cases after using ICL, as shown in Figure 3. It can be observed

Table 2: The performance of ICL with different demonstration selection methods using Llama3.1-8b. The best performance of each setting is marked in **bold**.

| Method | GSM8K | MATH | ARC-C | MMLU-Pro | FinQA | Amazon |
|---|---|---|---|---|---|---|
| Zero-Shot | 86.4 | 48.4 | 82.1 | 50.4 | 49.7 | 63.5 |
| BM25 (Robertson & Zaragoza, 2009) | 84.2 | 50.8 | 80.2 | 53.0 | 54.6 | 68.5 |
| GTR (Luo et al., 2023) | 82.1 | 50.8 | 80.5 | 53.5 | 55.0 | 68.5 |
| Yang et al. (2023) | 84.2 | 50.2 | 80.9 | 53.3 | 55.0 | 69.0 |
| Influence (Nguyen & Wong, 2023) | 83.9 | 51.0 | 81.3 | 52.4 | 54.6 | 69.5 |
| IDS (Qin et al., 2024) | 85.3 | 50.4 | 82.4 | 52.4 | 54.6 | 68.0 |
| Ours | **86.4** | **51.2** | **82.5** | **54.3** | **55.1** | **70.0** |

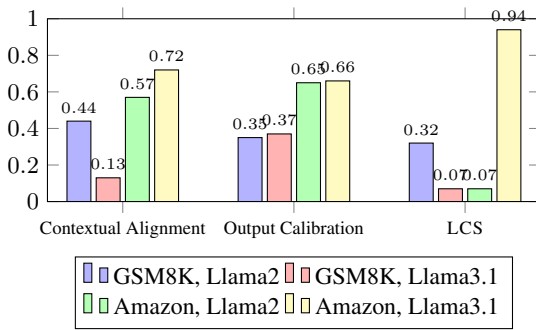

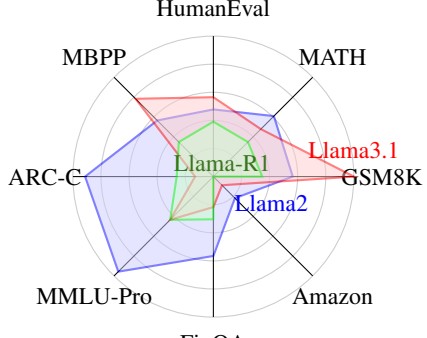

Figure 4: The results of the contextual alignment capability ($\hat{p}(D|Q)$), the output calibration capability ($\hat{p}(X|Q)$) and LCS of Llama2-7b and Llama3.1-8b on GSM8K and Amazon Review. $\hat{p}(D|Q)$ and $\hat{p}(X|Q)$ are calculated as the average value on all test data.

Figure 5: The intercept of the fitted line on each dataset and each model. We also compare the intercepts of Llama3.1 under different scales in Appendix F.4.

that: *(i)* Even on data where ICL does not improve performance, LCS still reveals the ICL effectiveness, proving the higher reliability of our metric compared with the performance-based metric. *(ii)* LCS is higher reliability, unlike performance which is susceptible to factors like the instruction, as it directly evaluates $p(X|Y)$ by using $Y$ as input and $X$ as output without relying on instructions (Appendix D.2), thus providing a more faithful reflection of the ICL effectiveness.

### 3.2.3 THE LEARNING GAIN IS A GOOD METRIC FOR DEMONSTRATION SELECTION

Enhancing ICL performance has been a topic of significant interest. Although this paper does not primarily focus on improving ICL performance, the discussions in §3.2.1 reveal several potential avenues for improvement. We observe that while there is a general positive correlation between the decrease in loss and the information learned from demonstrations, there also exist cases where demonstrations with rich information yield a low decrease in loss. To address this, we propose a method that first generates a preliminary answer $\hat{X}$ for the user question and then selects the demonstrations with a high learning gain. As shown in Table 2, our method outperforms other baselines, demonstrating the effectiveness of the learning gain-based method. In addition, on the datasets with low ICL effectiveness (e.g., GSM8K, ARC-Challenge), all methods have not brought significant improvement, which is consistent with our conclusion in §3.2.1.

### 3.3 RQ2. HOW DO DIFFERENT FACTORS INFLUENCE THE ICL EFFECTIVENESS?

### 3.3.1 THE MAIN FACTORS THAT INFLUENCE THE ICL EFFECTIVENESS

In §2.2, we discuss the main factors influencing the ICL effectiveness, including the contextual alignment capability ($\hat{p}(D|Q)$) and the output calibration capability ($\hat{p}(X|Q)$). In this section, we conduct experiments to analyze these conclusions further. We calculate the average values of $\hat{p}(D|Q)$ and $\hat{p}(X|Q)$ with Llama2-7b and Llama3.1-8b on GSM8K and Amazon Review, as shown in Figure 4.

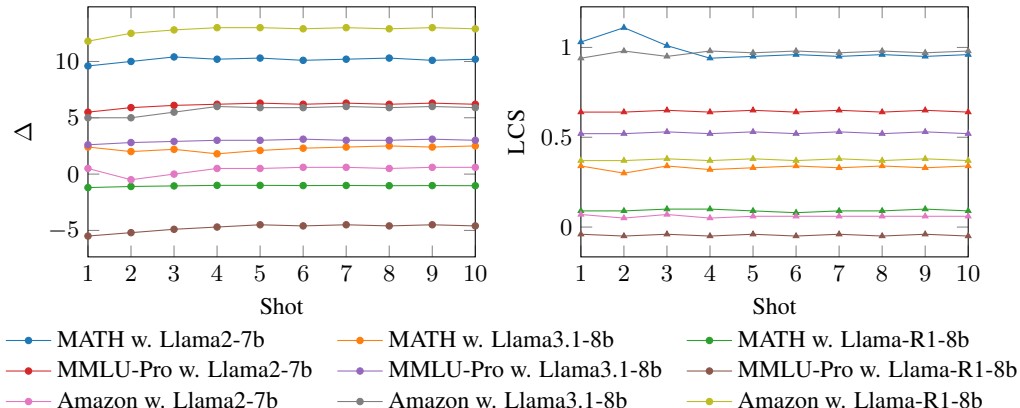

Figure 6: The performance change (Y-axis, left figure) and LCS (Y-axis, right figure) on MATH, MMLU-Pro, and Amazon Review with different shots (X-axis). The lines of the same color denote the results under the same setting.

From the figure, we can observe the following: *(i)* The results of Llama2-7b on Amazon Review indicate that the model is unable to effectively learn the information relevant to the user input from the provided demonstration $D$, *i.e.*, the contextual alignment capability is low, which leads to poor ICL effectiveness; *(ii)* The results of Llama3.1-8b on GSM8K show that although the ICL ability of the model is high, the model can accurately assess the relationship between input and output, *i.e.,* the output calibration capability also diminishes the ICL effectiveness; *(iii)* LCS is not equal to $\frac{\hat{p}(D|Q)}{\hat{p}(X|Q)}$, due to the error between $p$ and $\hat{p}$, as discussed in detail in Appendix C.3.

### 3.3.2 It is Harder for ICL to Improve the Learning Gain on Stronger Model

Apart from the slope, the intercept of the fitted line also reflects the effectiveness of ICL under different settings. We examine the intercept under different datasets and models, which is shown in Figure 5. From the figure, we observe that as model capacity increases, the corresponding intercepts decrease, indicating that: *(i)* From the perspective of the learning gain, the intercept reflects the overall magnitude of learning gain attributed to demonstrations for a given model and task, where a smaller intercept suggests less learning gain. *(ii)* From the perspective of error estimation (Appendix C.3), a smaller intercept implies a smaller discrepancy between $p$ and $\hat{p}$, meaning that the empirical predictor more closely approximates the oracle predictor. In summary, as model capacity increases, model predictions become more aligned with the oracle predictor, but the overall learning gain from demonstrations also diminishes.

### 3.3.3 More Shots Improve the ICL performance but not Effectiveness

To observe the differences in the ICL effectiveness under varying shot numbers, we conduct experiments with different shot numbers. Since Theorem 1 can calculate the influence of only a single demonstration, we divide the k-shot into k data points to calculate LCS. The experimental results are shown in Figure 6. From the figure, we can observe that: *(i)* As the number of shots increases, the overall performance change shows an upward trend. However, LCS does not generally increase or decrease with the number of shots but rather exhibits some degree of fluctuation. This is because the value of LCS is related to the inherent ICL effectiveness on a given model and dataset, while increasing the shot number cannot affect the ICL effectiveness. *(ii)* Relatively, the fluctuation of LCS gradually decreases as the number of shots increases. As discussed in Appendix C.3, increasing the number of shots can reduce computational errors, making the calculated result of LCS more stable and a more accurate reflection of the ICL effectiveness.

### 3.4 RQ3. Can Synthetic Data Accurately Reflect the ICL Effectiveness?

To verify the conclusions regarding the computation of LCS for synthetic data presented in §2.3, we conduct experiments to calculate LCS using synthetic data. During synthesis, we follow the procedure in Wang et al. (2025b) by inputting the task definition to generate corresponding demon-

Table 3: Performance change ($\Delta$) and LCS using labeled and synthetic demonstrations.

| Dataset | Type | Llama2-7b | | Llama3.1-8b | | Llama-R1-8b | |
|---|---|---|---|---|---|---|---|
| | | $\Delta$ | LCS | $\Delta$ | LCS | $\Delta$ | LCS |
| MATH | Labeled | +9.6 | 1.03 | +2.4 | 0.34 | −1.2 | 0.09 |
| | Synthetic | +6.0 | 0.75 | +1.3 | 0.16 | −1.8 | 0.05 |
| ARC-C | Labeled | +11.6 | 0.74 | −1.9 | −0.54 | −0.3 | 0.08 |
| | Synthetic | +8.2 | 0.53 | −1.5 | −0.56 | −0.2 | 0.06 |
| MMLU-Pro | Labeled | +5.5 | 0.64 | +2.6 | 0.52 | −5.5 | −0.04 |
| | Synthetic | +3.6 | 0.33 | +2.0 | 0.32 | −4.0 | −0.12 |
| Amazon | Labeled | +0.5 | 0.07 | +5.0 | 0.94 | +11.8 | 0.37 |
| | Synthetic | +0.0 | 0.0 | +4.0 | 0.42 | +9.0 | 0.25 |

strations. In each iteration, we provide the model with the task definition and the synthetic results from the previous iteration (empty in the first iteration), ask the model to generate demonstrations. The prompt we used is shown in Appendix D.3. We set the temperature to 0.9 and top_p to 0.9, sampling 8 demonstrations per iteration. A multi-round iterative process is used to ensure the diversity and quality of the synthesized demonstrations. Considering the computational resource limit, we only adapted experiments on four datasets, which are shown in Table 3. From the table, we observe the following: *(i)* The trend of LCS using synthetic data is consistent with that derived from labeled data, demonstrating that synthetic data can effectively reflect the ICL effectiveness. *(ii)* Compared to labeled data, the values of LCS obtained from synthetic data are relatively smaller, which supports the conclusion of Theorem 2.

## 4 RELATED WORK

**In-Context Learning** In-context learning guides the LLM reasoning process by providing several task-relevant demonstrations in the input, thereby improving performance (Brown et al., 2020; Dong et al., 2024; Zhao et al., 2025). Existing ICL research can be broadly categorized into two main areas: constructing high-quality demonstrations and improving demonstration selection performance. For demonstration construction, many works focus on the offline enhancement of demonstration quality. This includes methods aimed at increasing demonstration diversity, for instance, by generating synthetic data tailored to a given task or by selecting diverse demonstrations to improve compositional generalization (Wang et al., 2024a; 2025a; Chen et al., 2023a; Su et al., 2024; Levy et al., 2022). Another key aspect of offline construction is synthesizing or augmenting reasoning steps within existing demonstrations to better guide the inference process (Li et al., 2024; Zelikman et al., 2022; ZHAO et al., 2023). Other methods focus on the online synthesis of demonstrations, where demonstrations are generated or rewritten dynamically based on the user input to enhance reasoning performance, sometimes even leveraging the LLM itself to create these demonstrations (He et al., 2024; Chang & Fosler-Lussier, 2023; Kim et al., 2022). In the domain of demonstration selection, research primarily explores how to choose demonstrations most relevant to the user query, with some approaches also incorporating active learning principles to identify the most informative demonstration (Luo et al., 2024; Vu et al., 2023). Selection strategies include those based on n-grams (Li et al., 2023), semantic similarity using embeddings (Yang et al., 2023; Luo et al., 2023), or hybrid methods that combine multiple diverse strategies for retrieval and ranking (Wan et al., 2025; Wang et al., 2024c; Hao et al., 2022).

**Mechanism Analysis of In-Context Learning** Many studies have investigated the mechanisms underlying ICL for improving reasoning performance (Zhou et al., 2024; Dong et al., 2024). One line of research explores the mechanism of ICL by controlling the types of tasks used during pre-training (Edelman et al., 2024b; Han et al., 2023). Current mainstream work suggests that ICL ability arises from task diversity rather than data scale, with models gradually generalizing from solving in-domain tasks to solving out-of-domain tasks (Raventos et al., 2023). Additionally, some studies find that the modules responsible for knowledge acquisition and ICL ability are functionally independent (Nguyen & Reddy, 2025). Increasing the amount of data primarily enhances the knowledge-related components, while improvements in ICL depend more on the diversity of tasks encountered during training. Another line of work focuses on ICL reasoning, aiming to discover

the ICL mechanism by examining the relationship between provided demonstrations and the user question (Park et al., 2025; Li et al., 2025; Min et al., 2022; Wang et al., 2023a). Some studies argue that models perform ICL by learning the mapping between inputs and labels in the demonstrations, thereby improving task-solving performance (Kossen et al., 2024). Other research suggests that models learn the reasoning process embedded in the demonstrations and enhance reasoning performance by understanding and mimicking these processes (Lampinen et al., 2022).

However, the aforementioned studies mainly focus on improving the ICL performance or explaining the mechanism of ICL, often presuming that ICL is inherently effective. In contrast, recent studies have shown that ICL does not lead to performance improvement on certain tasks and models (DeepSeek-AI et al., 2025; Huang & Wang, 2025; Zheng et al., 2025). In this work, we investigate the main factors influencing the ICL effectiveness and propose the metric to evaluate the ICL effectiveness, to inform and inspire future research.

## 5 CONCLUSION

In this paper, we propose a novel metric LCS, to evaluate the ICL effectiveness. LCS overcomes the low reliability and poor attribution issues of performance-based metrics by measuring the variation in the learning gain with the contextual relevance. Based on LCS, we first discuss two primary factors that contribute to poor ICL effectiveness: poor contextual alignment capability and strong output calibration capability, demonstrating the strong attribution of LCS. Analytical experiments show that LCS can effectively reflect the effectiveness of ICL even on demonstrations where ICL does not lead to performance improvements, indicating high reliability. Furthermore, we present that the results of LCS on synthetic data are lower than those on real data, to inspire the application of LCS in data-insufficient scenarios.

## 6 REPRODUCIBILITY

We have provided all proofs of this paper in Appendix C.1, Appendix C.2 and Appendix C.4. We will release the experimental and pre-processed data and code upon the paper being accepted.

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

## A  LIMITATIONS AND ETHICS

### A.1  LIMITATIONS

*(i)* The current experimental datasets and models are limited, where future work will validate LCS on a broader range of models and datasets. *(ii)* Although we discuss that contextual alignment and output calibration capabilities are key factors influencing the ICL effectiveness, the underlying factors that affect these two capabilities warrant further investigation.

### A.2  ETHICS

All datasets and models used in this paper are publicly available, and our usage follows their licenses and terms. We employ AI tools for coding and writing polishing.

## B  LLM USAGE

We have employed the AI tool for coding and writing polishing.

## C  PROVE

### C.1  EQUATION 2

*Proof.* Suppose $X = (x_1, ..., x_{|X|})$, where $x_i$ is the token of $X$, we can derive that:

$$\mathbb{L}_p(X|K; D; Q) = -\log p(X|K; D; Q)$$

$$= \sum_{t=0}^{T} \left( -\log p(x_t|D; Q; x_{1:t-1}) \right)$$

$$= \sum_{t=0}^{T} \left( -\log \left( \frac{p(x_t|Q; x_{1:t-1})p(D|Q; x_{1:t})}{p(D|Q; x_{1:t-1})} \right) \right)$$

$$= \mathbb{L}_p(X|Q) - \sum_{t=0}^{T} \left( \log \left( \frac{p(D|Q; x_{1:t})}{p(D|Q; x_{1:t-1})} \right) \right)$$

$$= \mathbb{L}_p(X|Q) - (\log p(D|Q; X) - \log p(D|Q))$$

$\square$

### C.2  THEOREM 1

*Proof.*

$$p(X|Q; D) - p(X|Q) = \frac{p(X, Q, D)}{p(Q, D)} - \frac{p(X, Q)}{p(Q)}$$

$$= \frac{p(X, Q, D)p(Q) - p(X, Q)p(Q, D)}{p(Q, D)p(Q)}$$

$$= \frac{p(D|Q, X)p(Q, X)p(Q) - p(X, Q)p(D|Q)p(Q)}{p(Q, D)p(Q)}$$

$$= \frac{p(X|Q)}{p(D|Q)} \left( p(D|Q, X) - p(D|Q) \right)$$

$$= \frac{p(X|Q)}{p(D|Q)} I(X \to D|Q)$$

Therefore, we can conclude that:

$$I(X \to D|Q) = \frac{p(D|Q)}{p(X|Q)} I(D \to X|Q)$$

$\square$

### C.3 ERROR OF THEOREM 1

Assuming the error of the empirical predictor relative to the true predictor is $\hat{p}(A|B) = p(A|B) + \varepsilon(A|B)$, where $A, B$ are any random variables. We suppose that $r_p \geq \frac{\varepsilon(D|Q)}{\varepsilon(X|Q)} \geq \frac{\varepsilon(D|Q;X)}{\varepsilon(X|Q;D)}$, i.e., the error growth rate with introduced demonstrations is smaller than that without demonstrations, which is further smaller than the ICL effectiveness. According to Theorem 1, the slope of the fitted line can be approximated as:

$$\frac{I_{\hat{p}}(D|Q;X)}{I_{\hat{p}}(X|Q;D)} = \frac{(p(D|Q;X) - p(D|Q)) + (\varepsilon(D|Q;X) - \varepsilon(D|Q))}{(p(X|Q;D) - p(X|Q)) + (\varepsilon(X|Q;D) - \varepsilon(X|Q))}$$

Direct computation yields:

$$\frac{\hat{p}(D|Q)}{\hat{p}(X|Q)} = \frac{p(D|Q) + \varepsilon(D|Q)}{p(X|Q) + \varepsilon(X|Q)}$$

Thus, we have:

$$\Delta_I := \frac{I_{\hat{p}}(D|Q;X)}{I_{\hat{p}}(X|Q;D)} - \frac{I_p(D|Q;X)}{I_p(X|Q;D)} = \frac{\varepsilon(D|Q;X) - \varepsilon(X|Q;D)r_p}{I_p(X|D;Q)\left(I_p(X|D;Q) + \varepsilon(X|Q;D)\right)}$$

$$\Delta_p := \frac{\hat{p}(D|Q)}{\hat{p}(X|Q)} - \frac{p(D|Q)}{p(X|Q)} = \frac{\varepsilon(D|Q;X) - \varepsilon(X|Q;D)r_p}{p(X|D;Q)\left(p(X|D;Q) + \varepsilon(X|Q;D)\right)}$$

Assuming $I_p(X|D;Q) \leq p(X|D;Q)$, i.e., the information the model learns about $D$ from $X$ is less than the information inherently contained in the model, we have:

$$\Delta_I \leq \Delta_p$$

This implies that using the slope as a metric for ICL effectiveness has a smaller error compared to using $\frac{\hat{p}(D|Q)}{\hat{p}(X|Q)}$.

### C.4 THEOREM 2

*Proof.* Since $\hat{X} = \arg\max_{X \sim \mathcal{X}} \hat{p}(X|Q)$, we can conclude that $\hat{p}(\hat{X}|Q) \geq \hat{p}(X^*|Q)$. Based on the total probability theorem, we can draw that:

$$\hat{p}(\hat{D}|Q) = \sum_{X \sim \mathcal{X}} \hat{p}(\hat{D}|Q;X)\hat{p}(X)$$

$$\hat{p}(D^*|Q) = \sum_{X \sim \mathcal{X}} \hat{p}(D^*|Q;X)\hat{p}(X)$$

Considering that $\hat{p}(\hat{D}|Q;X) \leq \hat{p}(D^*|Q;X), \forall X \sim \mathcal{X}, Q \sim \mathcal{Q}$, it can be concluded that $\hat{p}(\hat{D}|Q) \leq \hat{p}(D^*|Q)$. Therefore, we can draw the conclusion that:

$$\frac{\hat{p}(\hat{D}|Q)}{\hat{p}(\hat{X}|Q)} \leq \frac{\hat{p}(D^*|Q)}{\hat{p}(X^*|Q)}$$

$\square$

## D ADDITIONAL INFORMATION

### D.1 DETAIL OF BENCHMARKS

In this section, we discuss the datasets we used in this paper in detail. The scale of the test set and the demonstrations of each dataset are shown in Table 4.

Table 4: The scales of test set and demonstrations of each dataset.

| Dataset | Test Set | Demonstration |
|---|---|---|
| GSM8K | 1319 | 7473 |
| MATH | 500 | 7496 |
| HumanEval | 164 | 596 |
| MBPP | 378 | 596 |
| ARC-Challenge | 1172 | 1119 |
| MMLU-Pro | 1000 | 70 |
| FinQA | 1147 | 6251 |
| Amazon Review | 200 | 1800 |

**GSM8K** GSM8K (Cobbe et al., 2021) is a high-quality dataset consisting of grade school level math problems. We directly use the training set as the demonstration pool.

**MATH** MATH (Hendrycks et al., 2021) is a dataset of high school competition-level math problems covering various domains, such as algebra, probability, and geometry. Following (Lightman et al., 2024), we use a sampled subset of $500$ examples for evaluation. We use the training set as the demonstration pool.

**HumanEval** HumanEval (Chen et al., 2021a) is a Python-based code generation benchmark. We follow the evaluation protocol of (Liu et al., 2023). Since the dataset does not provide a labeled training set, we use demonstrations from MBPP as the demonstration pool.

**MBPP** MBPP (Austin et al., 2021) is another Python-based code generation benchmark. Compared to HumanEval, it is larger in scale and includes a split between validation and test sets. In this paper, we adapt the evaluation on the test set and use the remaining data as the demonstration pool, following the evaluation protocol of (Liu et al., 2023).

**ARC-Challenge** ARC-Challenge (Yadav et al., 2019) is a difficult question-answering dataset focusing on scientific knowledge. We directly use the training set as the demonstration pool.

**MMLU-Pro** MMLU-Pro (Wang et al., 2024d) is a multitask benchmark designed to comprehensively evaluate LLMs on professional domain knowledge and complex reasoning capabilities. As the dataset only provides validation and test sets, we use the validation set as the demonstration pool and evaluate on the test set.

**FinQA** FinQA (Chen et al., 2021b) is a question-answering dataset in the financial domain. It requires models to perform numerical reasoning and calculations based on given financial tables and textual information. We use the training set as the demonstration pool.

**Amazon Review** The Amazon Review (Ni et al., 2019) dataset consists of numerous user ratings and textual reviews on products from the Amazon platform, and it is widely used in sentiment analysis and recommendation system research. Due to the large scale of the dataset, we select the *Health and Personal Care* category as the test set and use *All Beauty*, *Digital Music*, and *Software* as the demonstration pool.

### D.2    CALCULATION OF LCS

In this section, we present how to calculate LCS, which primarily involves two sequential steps: reasoning process paraphrasing and likelihood calculation. The prompts employed for these computations are detailed in Appendix D.3.

The reasoning process paraphrasing step requires models to restructure human-labeled reasoning processes according to their preferred reasoning style when provided with a given reasoning process. This adaptation is crucial because discrepancies between human-labeled reasoning formats and model-preferred reasoning patterns (e.g., "<think >" tag of Llama-R1 (DeepSeek-AI et al.,

2025)) could lead to inflated information gain measurements that reflect stylistic variations rather than knowledge acquisition. To mitigate this confounding factor, we implement reasoning process paraphrasing to eliminate format-induced biases, thereby ensuring that computational results authentically reflect knowledge-derived information learned from demonstrations. Specifically, for each data instance and demonstration, we input the question, answer, and human-labeled reasoning process (if provided), instructing the model to rephrase the output using its preferred reasoning style.

Following the paraphrasing, we calculate the likelihood with paraphrased results. For conditional probabilities expressed as $\hat{p}(A|B)$, we treat $B$ as user input and $A$ as model output, encapsulating them into a formatted string using the model chat template. This composite string is then processed through the model to obtain token-level likelihoods. The joint likelihood of a sequence $A$ is computed by multiplying the probabilities of all constituent tokens. To minimize the confounding effects of sequence length on probability comparisons, we apply length normalization to all computed likelihood values (Dai et al., 2019). This standardized approach ensures a fair comparison across outputs of varying lengths while preserving the probabilistic relationships between different reasoning processes.

### D.3 PROMPTS

In this section, we introduce the prompts used in this paper. The reasoning prompts of §3 can be seen in (Chen et al., 2023b; Grattafiori et al., 2024; DeepSeek-AI et al., 2025). The prompts used for the paraphrasing and the synthesis are shown in Table 5 and Table 6.

Table 5: The prompt of the paraphrase.

| Prompt of Paraphrasing |
| --- |
| < Begin of Task Definition > 
 {definition} 
 < End of Task Definition > 
 < Begin of Input > 
 {question} 
 < End of Input > 
 < Begin of Hint > 
 {hint} 
 < End of Hint > 
 < Begin of Answer > 
 {answer} 
 < End of Answer > 

 Considering the above task definition, generate the reasoning process of the given input and answer with the hint (could be empty). |

Table 6: The prompt of the demonstration synthesis.

| Prompt of Synthesis |
| --- |
| ```md 
 {task_definition} 
 ``` 

 Given Question: {question} 

 Based on the above task definition and the given question, synthesize a question and the corresponding answer that is similar to the given question of the task. |

# E ADDITIONAL DISCUSSION

## E.1 THE FACTORS THAT AFFECT ICL EFFECTIVENESS

Following the discussion of §2.2, in this section, we delve deeper into the factors that influence the ICL effectiveness, specifically the meaning of $\hat{p}(D|Q)$ and $\hat{p}(X|Q)$. Our primary focus is on different predictors $\hat{p}_1$ and $\hat{p}_2$ applied to the same data, assuming that the answer $X = \arg\max_{X \in \mathcal{X}} p(X|Q)$ is the correct answer, and the demonstration $D = \arg\max_{D \in \mathcal{D}} p(D|Q)$ is the most relevant demonstration to the question $Q$.

**Contextual Alignment Capability $\hat{p}(D|Q)$** If $\hat{p}_1(D|Q) \geq \hat{p}_2(D|Q)$, it indicates that $\hat{p}_1$ has a stronger ability to judge the relevance of demonstrations to the question compared to $\hat{p}_2$, showing that $\hat{p}_1$ is a better demonstration selector. From the perspective of demonstrations, this means that $\hat{p}_1$ is better at understanding the information in the demonstration and determining its relationship with the user question $Q$, reflecting that $\hat{p}_1$ has a stronger ICL ability than $\hat{p}_2$.

It is worth noting that while both $\hat{p}(D|Q)$ and $I(D \rightarrow X|Q)$ measure the consistency between the demonstration and the user input to some extent, their fundamental perspectives differ. $I(D \rightarrow X|Q)$ primarily focuses on the data perspective, measuring the relevance between the input and the demonstration under the assumption that the model is an oracle. In contrast, $\hat{p}(D|Q)$ primarily focuses on the model perspective, observing whether the model has the capability to gauge the relevance between the input and the demonstration, assuming that the demonstration is highly relevant to the user input.

**Output Calibration Capability $\hat{p}(X|Q)$** If $\hat{p}_1(X|Q) \geq \hat{p}_2(X|Q)$, it implies that $\hat{p}_1$ has a stronger ability to judge the correct answer compared to $\hat{p}_2$, meaning that $\hat{p}_1$ is a better answer scorer. It should be noticed that $\hat{p}(X|Q)$ does not directly reflect the model ability to solve the given question. This is because the model generates answers using greedy decoding, which means the generated answer could not be the answer with the highest likelihood. Rather, $\hat{p}(X|Q)$ represents the score the model assigns to a given answer, reflecting the model ability to assess the consistency between the answer and the question.

## E.2 EMPIRICAL THRESHOLD OF LCS

Specifically, based on Figure 2, we can use LCS = 0.2 as an empirical threshold, since when LCS $\leq 0.2$, the corresponding performance gain is minimal or negative, suggesting that ICL is less effective in the given task and model. This threshold is largely empirical. In practice, users can adjust the sensitivity to ICL effectiveness according to their preferences.

## E.3 EFFICIENCY OF LCS CALCULATION

LCS requires calculating four related likelihoods for each data point. Therefore, if we assume the time cost for a model to run one pass of ICL inference on a given dataset is $T$, the time cost of our method is $4T$. Although our time cost is greater than that of a single inference pass, our primary motivation is to propose an effective method for measuring ICL effectiveness to guide subsequent demonstration annotation and ICL usage, rather than to perform efficient inference. Therefore, we consider the additional time cost to be acceptable.

## E.4 WHY OUR ANALYSIS IS BASED ON CONDITIONAL PROBABILITY

We acknowledge that a growing body of work has demonstrated that the order of demonstrations in the prompt can affect the outputs of current autoregressive LLMs (Lu et al., 2022; Guo et al., 2024). In this paper, however, we deliberately work with an idealized order-invariant model in which the relevant conditionals are unaffected by permutations of the demonstrations. Prior research suggests that the observed order sensitivity is a limitation of existing models and training procedures, and that an ideal language model should, in principle, be invariant to the ordering of demonstrations (Xiang et al., 2024). Furthermore, several studies indicate that the influence of text order on in-context learning is diminishing as models and training improve (Pham et al., 2025), and many influential analyses of ICL adopt a similar order-invariance assumption when deriving related decompositions (Zhang

et al., 2025; Jesson et al., 2025). Under this widely used idealization, Equation 2 is well-defined and yields a decomposition of the loss into a "zero-shot" term and a "demo-dependent" term. We explicitly adopt this assumption in our derivation in Appendix C.1 to isolate the fundamental nature of ICL effectiveness and to obtain a clean measure of it. Our experimental results are consistent with the theoretical predictions obtained under this assumption, supporting the practical reasonableness of applying Equation 2 even when instantiated with standard autoregressive LLMs.

In this work, we define the predictive probability for an output sequence $\mathbf{X} = (x_1, \dots, x_T)$ as the product of token-wise conditional probabilities:

$$P_\theta(\mathbf{X} \mid \mathbf{Q}) = \prod_{t=1}^{T} P_\theta(x_t \mid x_{<t}, \mathbf{Q}).$$

This construction is widely used in the NLP research (Bengio et al., 2003; Sutskever et al., 2014), which follows directly from the chain rule of probability: the joint distribution of any discrete sequence can be decomposed, without loss of generality, into a left-to-right product of conditional distributions. Hence, as long as the model provides reasonable estimates of each conditional distribution $P_\theta(y_t \mid y_{<t}, \mathbf{x})$, the above product corresponds to the likelihood of the sequence under the model and can be naturally interpreted as its predictive probability, or a joint score.

### E.5 Why Removing Logarithm in Learning Gain

In this section, we discuss why, when defining the learning gain $I_p(X \to D \mid Q)$, we remove the log based on Equation 2. This is because, by using the difference in probabilities, we can better analyze its relationship with other factors. We consider this a reasonable transformation, since the transformed expression still reflects the change in loss, which is consistent with our motivation of measuring the effectiveness of ICL via loss change. Specifically, let $k = \log p(D \mid Q; X) - \log p(D \mid Q)$, i.e., the decrease in loss brought by introducing the demonstration. Then we have:

$$p(D \mid Q; X) = e^k p(D \mid Q)$$
$$\Rightarrow p(D \mid Q; X) - p(D \mid Q) = (e^k - 1)p(D \mid Q)$$

Since $p(D \mid Q) > 0$, $p(D \mid Q; X) - p(D \mid Q)$ is positively correlated with $\log p(D \mid Q; X) - \log p(D \mid Q)$. That is, $I_p(X \to D \mid Q)$ is positively correlated with the change in loss.

## F Additional Experiments

### F.1 Main Experiment Results

Table 7: The performance of different models on different datasets using 0-shot and 1-shot. $\Delta$ is the performance change of 1-shot compared with 0-shot. HumanE denotes HumanEval, A-C denotes ARC-Challenge, M-P denotes MMLU-Pro, and Amazon denotes Amazon Review.

| Model | Shot | Math | | Code | | Reason | | Domain | |
|---|---|---|---|---|---|---|---|---|---|
| | | GSM8K | MATH | HumanE | MBPP | A-C | M-P | FinQA | Amazon |
| Llama2-7b | 0 | 12.7 | 5.0 | 14.0 | 23.0 | 34.6 | 14.1 | 10.5 | 28.5 |
| | 1 | 27.7 | 14.6 | 13.4 | 22.5 | 46.2 | 19.6 | 17.8 | 29.0 |
| | $\Delta$ | +10.0 | +9.6 | −0.6 | −0.5 | +11.6 | +5.5 | +7.3 | +0.5 |
| Llama3.1-8b | 0 | 86.4 | 48.4 | 65.9 | 54.8 | 82.1 | 50.4 | 49.7 | 63.5 |
| | 1 | 84.2 | 50.8 | 63.4 | 55.6 | 80.2 | 53.0 | 54.6 | 68.5 |
| | $\Delta$ | −1.8 | +2.4 | −2.5 | +0.8 | −1.9 | +2.6 | +4.9 | +5.0 |
| Llama-R1-8b | 0 | 86.1 | 75.4 | 70.7 | 67.2 | 84.8 | 58.2 | 45.2 | 53.5 |
| | 1 | 80.1 | 74.2 | 67.7 | 60.8 | 84.5 | 52.7 | 43.4 | 65.0 |
| | $\Delta$ | −6.0 | −1.2 | −3.0 | −6.4 | −0.3 | −5.5 | −1.8 | +11.5 |

**Overall Performance** In this part, we present the performance of 0-shot and 1-shot on different models and datasets, as shown in Table 7.

**Figurative Illustrations of Learn-to-Context Slope**   In this section, we present the variation of $I_{\hat{p}}(X \to D|Q)$ with respect to $I_{\hat{p}}(D \to X|Q)$ under different settings, as illustrated in Figure 7, Figure 8, and Figure 9. Considering that the number of data points could vary slightly across different models due to the potential for excessively long responses from certain models (e.g., Llama-R1-8b could persist in "thinking").

## F.2   DIFFERENT SIMILARITY MEASUREMENT

In this section, we discuss the impact of replacing the contextual relevance $I_{\hat{p}}(D \to X|Q)$ with other metrics. We conduct experiments on Llama3.1-8b using the GSM8K, MATH, and Amazon Review dataset, where we replace the similarity measure with n-gram (Broder et al., 1997), BM25 (Robertson & Zaragoza, 2009), and cosine similarity (Singhal & Google, 2001) to evaluate the similarity between the provided demonstration and user input. The experimental results are shown in Figure 10, Figure 11, abd Figure 12. From the figure, we observe the following: *(i)* For effective similarity measures (e.g., BM25, cosine similarity), the observed ICL effectiveness is consistent with using the contextual relevance; *(ii)* However, for metrics with poorer performance (e.g., n-gram), the ICL effectiveness is not accurately reflected, demonstrating that n-gram fails to properly capture the similarity between demonstrations and user inputs.

## F.3   DIFFERENT MODEL

Table 8: Performance and fitted lines across different models and datasets. ARC-C denotes ARC-Challenge, and Amazon denotes Amazon Review. $\Delta$ denotes the performance change of 1-shot relative to 0-shot, where performance gains $< 1.0$ are marked in red. $r_{\hat{p}}x + b$ represents the fitted line with $I_{\hat{p}}(X \to D|Q)$ as the x-axis and $I_{\hat{p}}(D \to X|Q)$ as the y-axis, where $r_{\hat{p}}$ values $< 0.2$ are highlighted in red.

| Model | MATH | | FinQA | | Amazon | |
|---|---|---|---|---|---|---|
| | $\Delta$ | $r_{\hat{p}}x + b$ | $\Delta$ | $r_{\hat{p}}x + b$ | $\Delta$ | $r_{\hat{p}}x + b$ |
| Llama3.1-8b | +2.4 | $0.34x - 0.00$ | +4.9 | $0.82x - 0.06$ | +5.0 | $0.94x - 0.09$ |
| Llama3.1-70b | −3.6 | $-0.13x - 0.04$ | +7.6 | $0.77x - 0.07$ | +16.0 | $0.79x - 0.18$ |
| Qwen2.5-7b | +2.2 | $0.81x - 0.16$ | +4.9 | $0.29x - 0.09$ | +27.0 | $0.42x + 0.00$ |
| Qwen3-8b | −1.4 | $-0.21x - 0.08$ | +6.7 | $0.53x + 0.21$ | −1.0 | $0.03x - 0.13$ |
| Ministral-8b | −1.8 | $0.04x - 0.02$ | +4.8 | $0.71x - 0.04$ | +4.0 | $0.69x - 0.11$ |

To evaluate the effectiveness of LCS on the models with different scales and series, we adapt the experiments on Qwen2.5-7b (Qwen et al., 2025), Qwen3-8b (Yang et al., 2025), Ministral-8B-Instruct-2410 (Ministral-8b) (Jiang et al., 2023), and Llama3.1-70b (Grattafiori et al., 2024). The experimental results are shown in Table 8. It can be seen that LCS still reflects the ICL effectiveness on the models with different scales and series, proving the generalization of our metric.

## F.4   INTERCEPT UNDER DIFFERENT MODEL SCALES

To more thoroughly compare the differences in the effective information learned from demonstrations by LLMs of varying capabilities, we conduct experiments on LLMs of different scales within the same series. The experimental results are shown in Figure 13. From the figure, it can be observed that the intercept of Llama3.1-70b is generally smaller than that of Llama3.1-8b, as discussed in Section §3.3.2, indicating that Llama3.1-70b learns less effective information.

## F.5   PERFORMANCE OF LCS WITH MISMATCH LABEL

In this section, we investigate the effectiveness of LCS under adversarial labels. We mainly conduct experiments on the MATH and MMLU-Pro datasets. For MATH, following Madaan et al. (2023), we replace the values in the examples with placeholders. For MMLU-Pro, we randomly replace the choice corresponding to each question with another choice. The experimental results are shown in Table 9. From the table, we can see that when using adversarial labels, LCS can still faithfully reflect the effectiveness of ICL, which is consistent with prior work that adversarial labels can also lead to

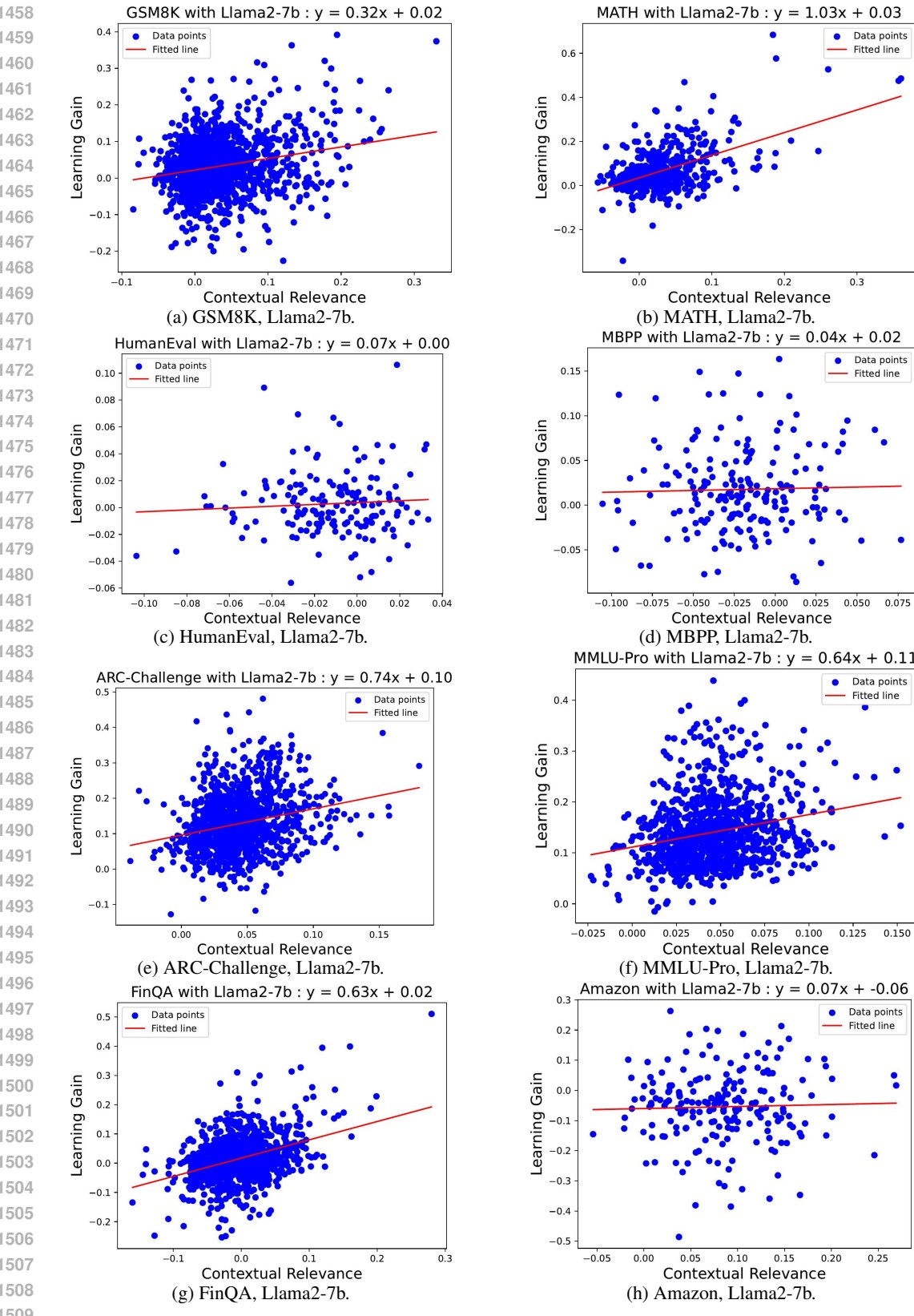

Figure 7: The variation of $I_{\hat{p}}(X \rightarrow D | Q)$ (y-axis) with $I_{\hat{p}}(D \rightarrow X | Q)$ (x-axis) on different datasets using Llama2-7b. The title of each plot displays the corresponding dataset and fitted line. Each blue dot in the plot represents a data point, and the red line indicates the fitted line of the data points.

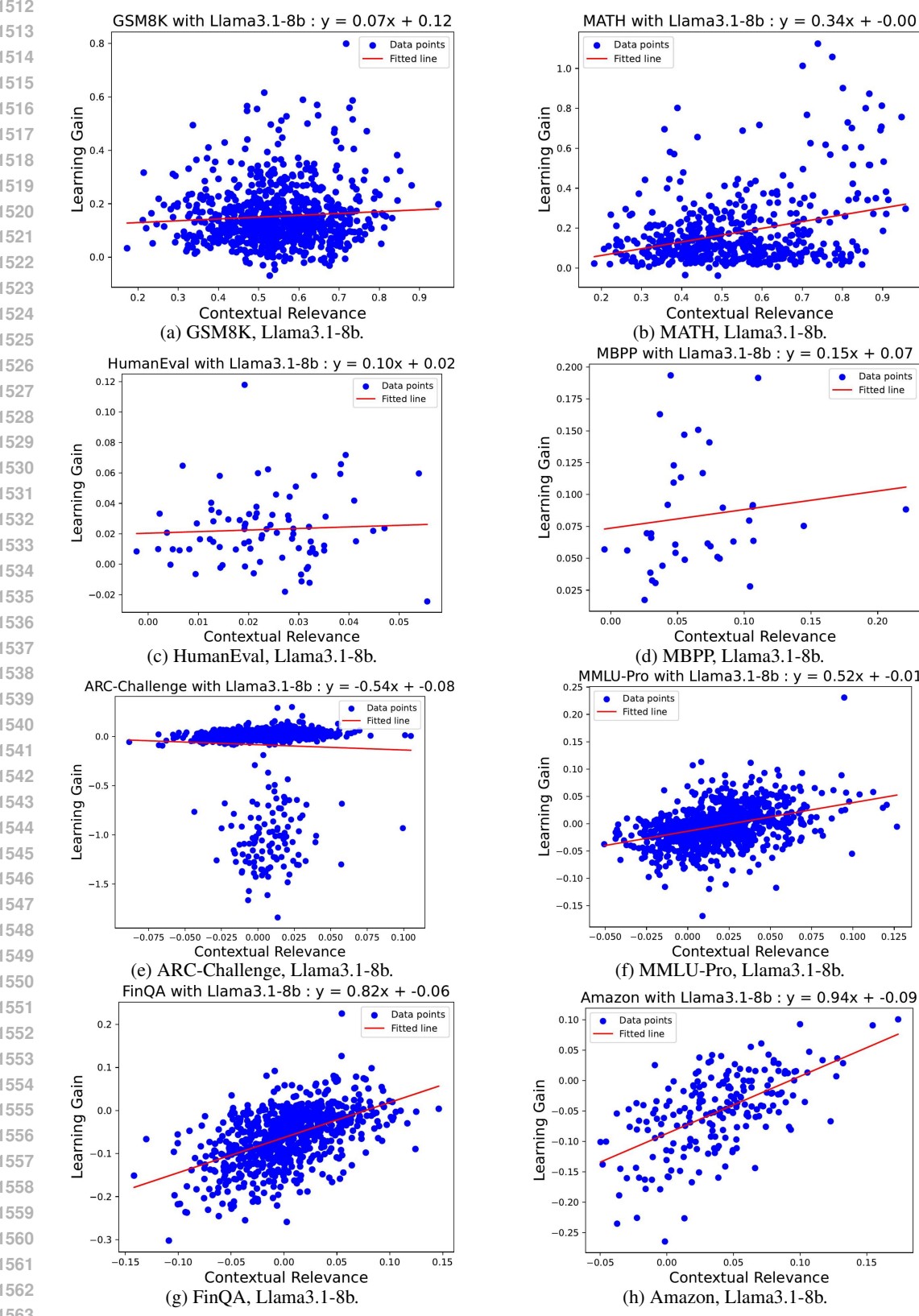

Figure 8: The variation of $I_{\hat{p}}(X \rightarrow D|Q)$ (y-axis) with $I_{\hat{p}}(D \rightarrow X|Q)$ (x-axis) on different datasets using Llama3.1-8b. The legend is the same as Figure 7.

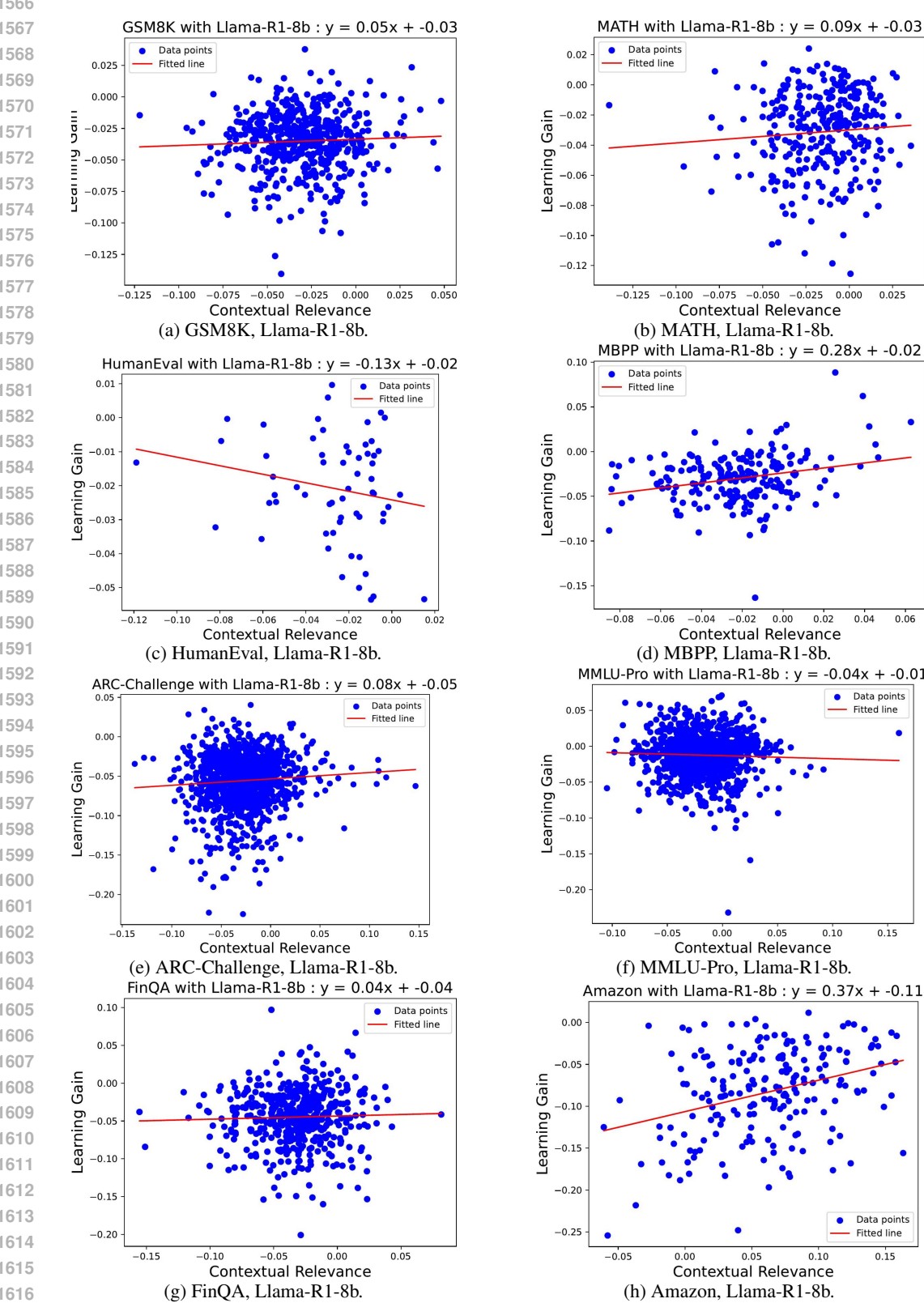

Figure 9: The variation of $I_{\hat{p}}(X \to D|Q)$ (y-axis) with $I_{\hat{p}}(D \to X|Q)$ (x-axis) on different datasets using Llama-R1-8b. The legend is the same as Figure 7.

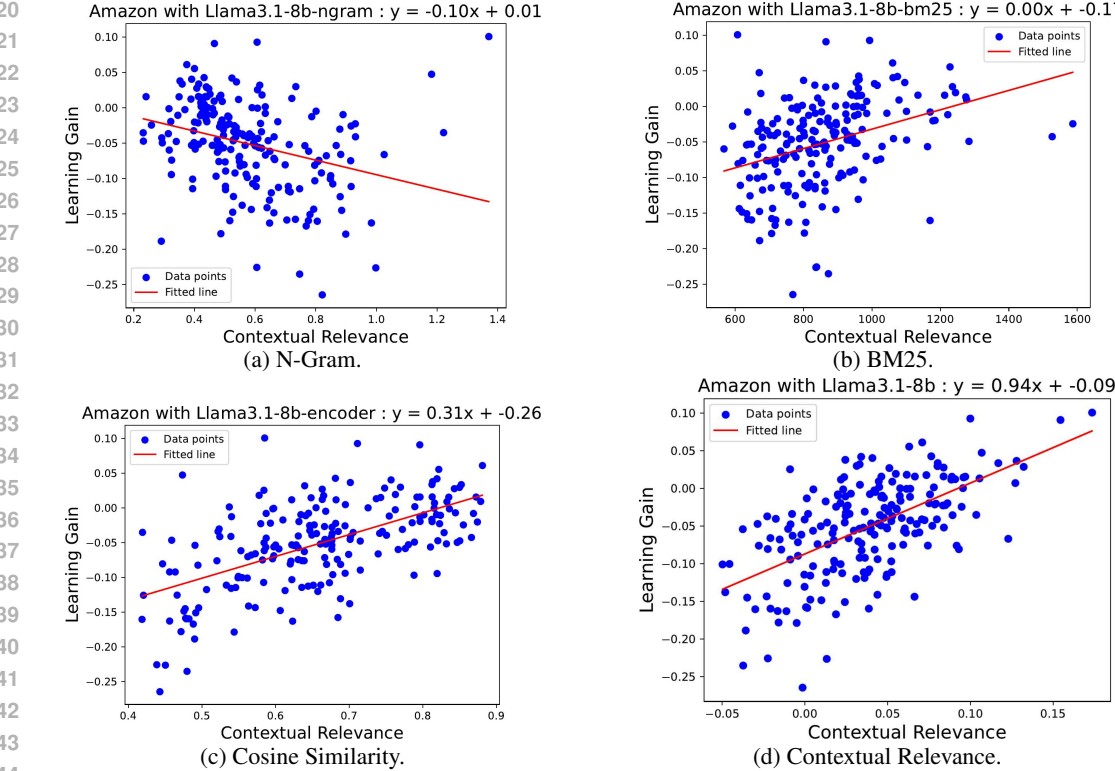

Figure 10: The variation of the learning gain (y-axis) with different similarity metrics (x-axis) on Amazon using Llama3.1-8b.

| Dataset | Type | Llama2-7b $\Delta$ | Llama2-7b LCS | Llama3.1-8b $\Delta$ | Llama3.1-8b LCS | Llama-R1-8b $\Delta$ | Llama-R1-8b LCS |
|---|---|---|---|---|---|---|---|
| MATH | Real | +9.6 | 1.03 | +2.4 | 0.34 | −1.2 | 0.09 |
| | Mismatch | +4.2 | 0.72 | +2.0 | 0.22 | −1.4 | 0.03 |
| MMLU-Pro | Real | +5.5 | 0.64 | +2.6 | 0.52 | −5.5 | −0.04 |
| | Mismatch | +4.3 | 0.52 | +1.9 | 0.30 | −5.0 | −0.12 |

Table 9: The performance with origin and adversarial labels. Real denotes the origin label, and Mismatch denotes the adversarial label.

performance improvements (Madaan et al., 2023). Moreover, compared with original labels, LCS under adversarial labels are relatively lower, which is consistent with the conclusion of Theorem 2, since demonstrations with adversarial labels are of lower quality than those using the original labels.

## F.6 LCS with Black-Box LLMs

| Model | GSM8K $\Delta$ | GSM8K $r_{\hat{p}}x + b$ | MATH $\Delta$ | MATH $r_{\hat{p}}x + b$ | FinQA $\Delta$ | FinQA $r_{\hat{p}}x + b$ |
|---|---|---|---|---|---|---|
| gpt-5-nano | −1.1 | $0.01x + 0.27$ | −3.9 | $−0.12x − 0.03$ | +7.6 | $0.31x + 0.05$ |

Table 10: LCS with gpt-5-nano on GSM8K and MATH. The probability is generated following Kaneko et al. (2025). Considering the API cost, we randomly sample 128 examples from each dataset.

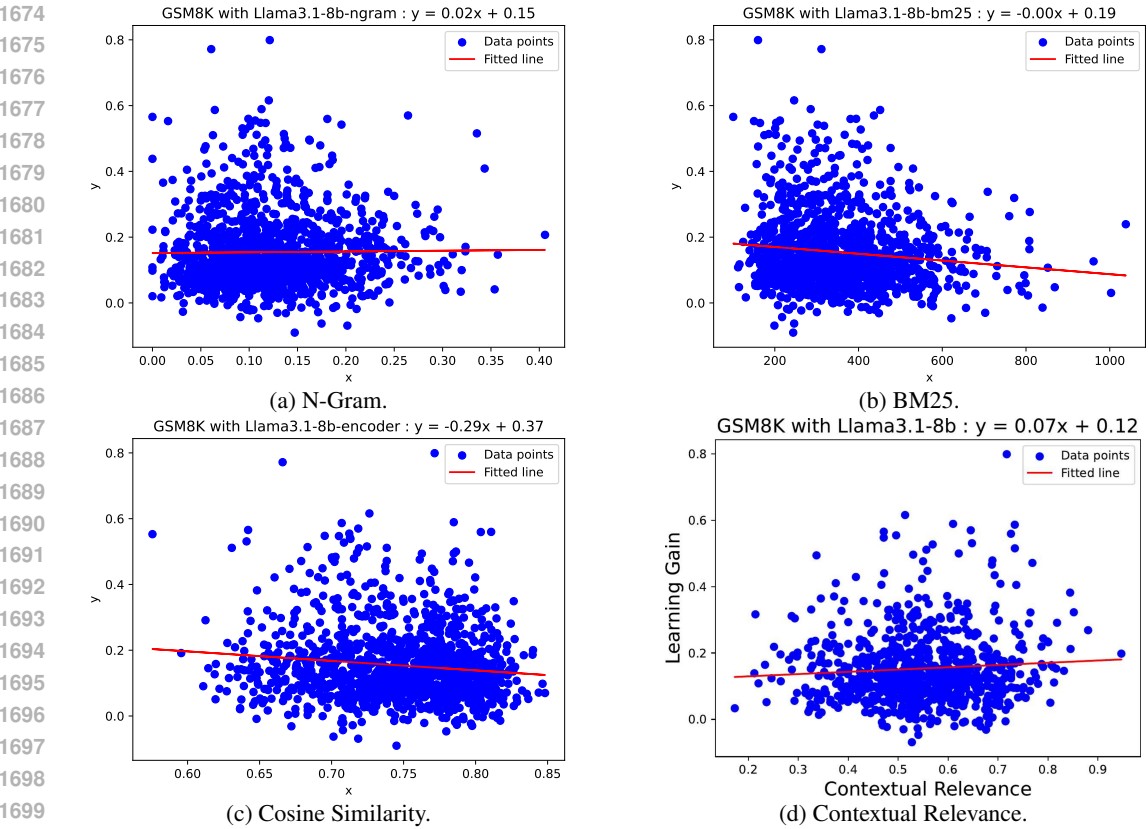

(a) N-Gram.

(b) BM25.

(c) Cosine Similarity.

(d) Contextual Relevance.

Figure 11: The variation of the learning gain (y-axis) with different similarity metrics (x-axis) on GSM8K using Llama3.1-8b.

Based on Theorem 1, the main point to calculate LCS with black-box LLMs is how to obtain the logprob. Following Lee et al. (2023); Kaneko et al. (2025), we employ a sampling-based pseudo-likelihood estimator for recovering LLM output distributions from samples. Specifically, for each prompt–response pair, we first feed the prompt into the model and sample multiple outputs. Then, we compute the average ROUGE-N score between these outputs and the response. Kaneko et al. (2025) shows that when the number of samples is sufficiently large, this average ROUGE-N score converges to the real probability. Although the above computation procedure is not very efficient, here we only provide an idea of how to apply LCS to black-box models, and improving the efficiency of computing log probabilities is beyond the scope of this paper.

We conduct experiments using `gpt-5-nano` (OpenAI, 2025). Due to the high API cost, we only randomly sample 128 examples from GSM8K and MATH for our experiments. The experimental results are shown in Table 10, from which we can observe that LCS also accurately reflects the effectiveness of ICL, thereby demonstrating the feasibility of applying LCS to black-box models.

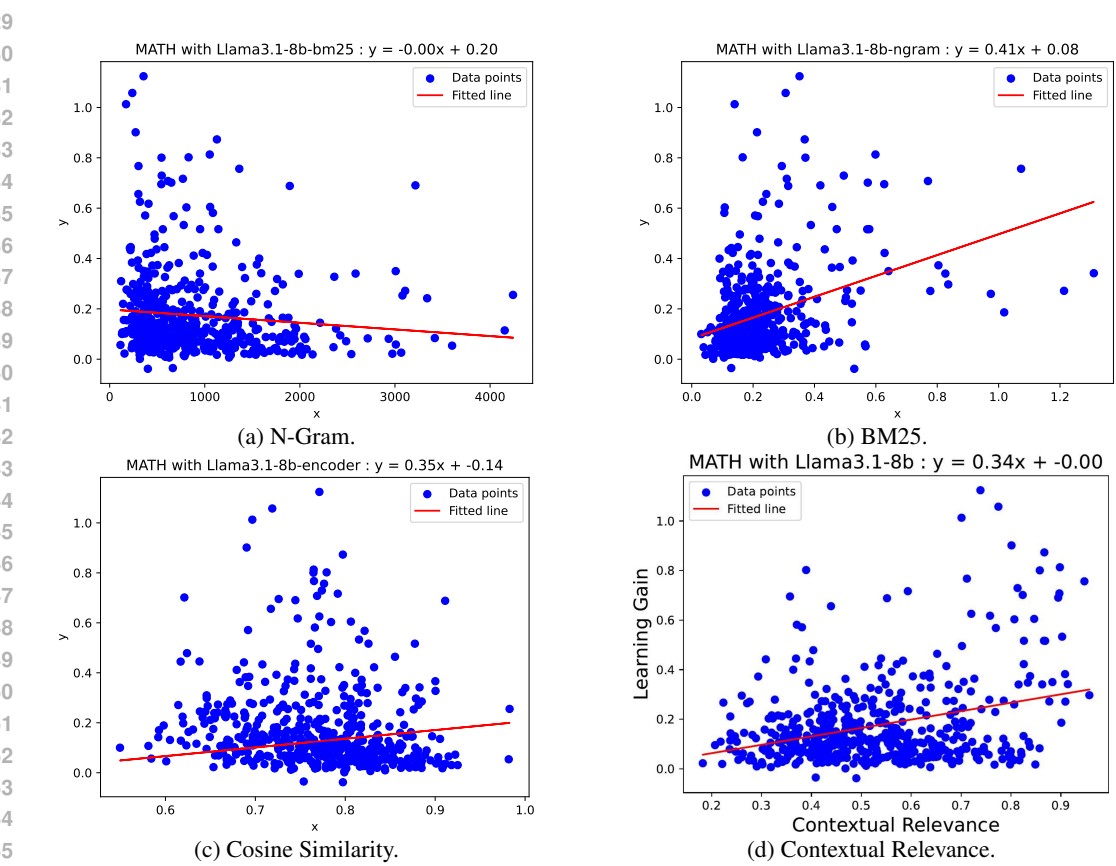

Figure 12: The variation of the learning gain (y-axis) with different similarity metrics (x-axis) on MATH using Llama3.1-8b.

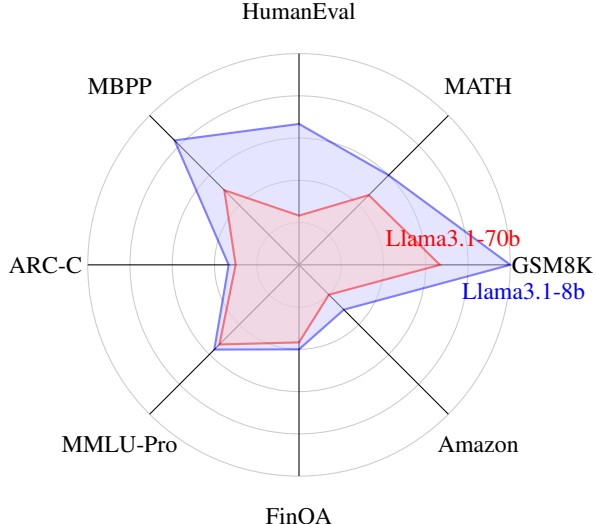

Figure 13: The intercepts of the fitted lines of Llama3.1-8b and Llama3.1-70b.

