# OpenReview forum: "Learning-to-Context Slope: Evaluating In-Context Learning Effectiveness Beyond Performance Illusions"
_ICLR.cc/2026/Conference — ICLR 2026 Conference Desk Rejected Submission_

### Official Review · Reviewer_yHFq · 2025-11-01

**Soundness:** 3
**Presentation:** 2
**Contribution:** 2
**Rating:** 4
**Confidence:** 3

**Summary:**

The paper introduces Learning-to-Context Slope (LCS) to measure the in-context learning (ICL) effectiveness of the language models. Through empirical evaluations, authors show that LCS is more reliable compared to the existing evaluation method, namely model performance change with respect to demonstrations.

**Strengths:**

- Author provides mathematical formulation of LCS, and also provides intuitive analysis on its components.
- LCS shows promising empirical results compared to the existing metrics.

**Weaknesses:**

- Although authors provide mathematical formulation of LCS, the motivation of choosing LCS to overcome the problems of existing evaluation metric is not clear.
- The purpose of 2.3 is not clear. Since they empirically show that LCS can be effective even with synthetic data, I do not think this section is necessary. To me discussing about real large language models, which they placed in the appendix might have been more informative.
- Organization of Section 3 is not optimal. The limitation of the existing evaluation metric is (1) low reliability (2) poor attribution (3) requirement of labeled data. Then the experiments should first deal with these aspects of LCS.
- The implications of Section 3.2.1 is not clear.
- LCS cannot be applied to black box models.

**Questions:**

Refer to the weaknesses.

---

> ### Author Response · Authors · 2025-11-16
>
> Thank you for your positive assessment of our work! **We have revised the paper according to your review; all changes are highlighted in blue**. Below, we respond to your specific comments:
>
> *1. Although the authors provide a mathematical formalization of LCS, their motivation for choosing LCS to address the problems of existing evaluation metrics is not clear. (**Lines 83-93**)*
> * **In Lines 83-93, we explain how LCS overcomes the limitations of existing metrics**:
>   * LCS measures ICL effectiveness using loss, which yields higher reliability than performance-based metrics.
>   * LCS can quantify the impact of different factors on ICL effectiveness and therefore provides better attribution.
>   * We show how LCS changes when using synthetic data, thereby removing the dependence on human-annotated data.
>
> 2. *Section 2.3 is unnecessary. Moving the discussion of real large language models from the appendix to the main text would be more informative. (**Section 2.3 in Revision**)*
> * Thank you for your suggestion! **We have revised the theorem statements in Section 2.3 so that they extend to comparing demonstrations of different qualities when evaluating the effectiveness of LCS**, thereby enhancing the value of this section.
> * Since the appendix contains an extensive discussion of real large language models, we would be happy to move relevant parts into the main text if you could indicate which specific paragraphs you are referring to!
>
> 3. *The organization of Section 3 is not ideal.*
> * Our current Section 3 is already organized in the order you suggested, and **the titles of Sections 3.2, 3.3, and 3.4 directly correspond to the limitations of existing evaluation metrics**. Specifically:
>   * RQ1 in Section 3.2 discusses the reliability of different metrics and shows that ICL is more reliable than existing performance-change-based metrics.
>   * RQ2 in Section 3.3 examines different factors that influence ICL effectiveness, demonstrating that LCS can be used to attribute poor ICL effectiveness.
>   * RQ3 in Section 3.4 investigates measuring ICL effectiveness on synthetic data, showing that LCS can accurately reflect ICL effectiveness without requiring labeled data.
>
> 4. *The meaning/implications of Section 3.2.1 are not clear.*
> * The implications of Section 3.2.1 **are exactly the boldfaced sentences at the beginning of each paragraph**, including:
>   * We analyze that ICL effectiveness is not strongly correlated with dataset difficulty or model capability.
>   * In cases where performance improvement can reflect ICL effectiveness, our method can also indicate ICL effectiveness.
>
> 5. *LCS cannot be applied to black-box models. (**Appendix F.6 in Revision**)*
> * In the revision, we have added an explanation of how to apply LCS to black-box models, namely, **we first fit the likelihood of the black-box model’s generation probabilities and then use this fitted likelihood to compute LCS**.
> * Following the method of [1], we approximately obtain the log probabilities of a black-box model and conduct experiments using gpt-5-nano. Considering the API cost, we only use 128 randomly sampled examples from GSM8K and MATH for our experiments. The results are shown in the table below. We can observe that LCS also accurately reflects the effectiveness of ICL, thereby demonstrating the feasibility of applying LCS to black-box models.
>
> | **Model**    | GSM8K $\Delta$ | GSM8K $r_{\hat{p}}x + b$ | MATH $\Delta$ | MATH $r_{\hat{p}}x + b$ | FinQA $\Delta$ | FinQA $r_{\hat{p}}x + b$ |
> |-------------|----------------|--------------------------|---------------|-------------------------|----------------|--------------------------|
> | gpt-5-nano  | $-1.1$         | $0.01x + 0.27$           | $-3.9$        | $-0.12x-0.03$           | $+7.6$         | $0.31x+0.05$             |
>
> [1] Sampling-based Pseudo-Likelihood for Membership Inference Attacks. ACL 2025 Findings.

---

> ### Author Response · Authors · 2025-11-28
>
> Dear Reviewer, we have revised the paper based on your review and submitted a response, and we sincerely hope that our changes address your concerns. As there are now fewer than five days remaining before the rebuttal period ends, we would be truly grateful for your feedback and sincerely look forward to your reply.

---

### Official Review · Reviewer_bnZL · 2025-11-02

**Soundness:** 1
**Presentation:** 2
**Contribution:** 2
**Rating:** 2
**Confidence:** 5

**Summary:**

This paper comes up Learning-to-Context Slope (LCS), a novel metric that quantifies the effectiveness of in-context learning (ICL). LCS is defined as the slope between **learning gain** $I_p(X \to D\mid Q)=p(D\mid Q;X)-p(D\mid Q)$ and **contextual relevance** $I_p(D \to X\mid Q)=p(X\mid Q;D)-p(X\mid Q)$ where $p$ is the predictive distribution, $Q$ is the user input, $D$ is the ICL demonstration (e.g., one-shot example), and $X$ is the labeled output. Such a metric seems the most useful when one does not have many labeled examples to examine the actual ICL accuracies (= few-shot - one-shot). Furthermore, the authors show that creating synthetic demonstrations and then applying LCS is well-correlated with ICL effectiveness.

Contributions:
- Proposes LCS as a metric to measure ICL effectiveness and shows strong correlation with task performance improvement with ICL over 8 datasets with multiple models. Theorem 1 claims these are linearly related with slope $p(D\mid Q)/p(X\mid Q)$ and the paper provides Gives an empirical procedure to estimate the slope via least-squares on sampled triples of $(q_i, x_i, d_i)$.
- Explores “label-free” estimation with synthetic demonstrations and argues such LCS is consistently smaller than from real data (Theorem 2).
- Identifies two factors claimed to modulate LCS: contextual-alignment capability $\hat{p}(D∣Q)$ and output-calibration capability $\hat{p}(X∣Q)$.
- Provides an empirical LCS threshold ≈ 0.2 and a heuristic for ICL demo selection based on learning gain, which achieves strong performance compared to other baseline selection methods.

**Strengths:**

- Originality: Framing ICL effectiveness as a slope between two “information-like” quantities is novel and produces a continuous signal where EM/Pass@1 are binary/noisy.
- Clarity: The empirical estimator (Eq. 3) and plotting protocol are easy to reproduce conceptually, and the paper is upfront that $r_{\hat{p}}= \hat{p}(D∣Q) / \hat{p}​(X∣Q)$ is errorful. (however, see weaknesses and questions for multiple points of unclear presentation.)
- Quality: Broad evaluation across 8 datasets on several model families and choosing ICL examples based on their heuristic is particularly helpful.
- Significance: If sound, LCS could help decide when to invest in demonstrations and guide demo selection when labeling comes at a high cost.

**Weaknesses:**

1. **Foundational mismatch between Eq. (2) and autoregressive LLMs**: The proof underlying Eq. (2) treats conditionals like $p(D\mid Q;X)$ and $p(D\mid Q)$ as if the order of variables in the conditioning can be freely rearranged; this yields a neat decomposition of loss into a “zero-shot” term and a “demo-dependent” term (Eq. 2). Yet in an autoregressive LLM, $\log p(x_t \mid Q, D, x_{<t})$ depends on the exact **prompt order** $[Q, D, x_{<t}]$. Specifically, for an AR LM, $p(\cdot\mid\cdot)$ is implemented as **conditional next-token probabilities over a specific prompt order**. Because the paper’s derivation ignores this fixed ordering, Eq. (2) does not work if one just plugs in an likelihood by LLM into $p$. The proof in Appendix C.1 must assume that an order-invariant joint over $(Q,D,X)$. So Eq. (2) does not hold for standard AR LLM $p$ that the paper later uses unless we assume order invariance, which is an extremely strong assumption on language modeling.

2. **Strong assumptions needed for theoretical results to support LCS**:
- **Pseudo-probabilities that do not align with theorems**: In practice the method forms a chat-template string, multiplies token probabilities to get a joint score, and then applies length normalization “to minimize the confounding effects of sequence length,” which is not a proper probability transformation and severs the direct connection to its mathematical justifications.
- **Theorem 2 assumes a universally optimal demonstration $D^\star$.**: Theorem 2 assumes a real demonstration $D^\star$ such that for all $X, Q$ s.t. $\hat p(D^\star \mid Q;X)$ dominates $\hat p(\hat D\mid Q;X) $. In plain English this means that there exists a single demonstration that is uniformly best across questions and answers, which seems implausible. The assumption is doing heavy lifting for the conclusion, unless the authors can empirically demonstrate such an assumption is not strong.
- **Mismatch between the loss motivation and the probability‑difference metric**: The loss motivation is in logarithms (Eq. 2), but the core quantities are later defined as differences of probabilities, and the slope result relates $I_p$ terms linearly. I'm confused what Section 2.1 is trying to show then.
3. **LCS reflects ICL effectiveness are only partially substantiated**:
- **Weak baselines for demo selection**: When proposing learning-gain–based selection, the comparison set is limited (BM25, GTR, IDS), omitting many modern ICL selection strategies (e.g., stronger LLM-scoring/uncertainty-aware retrievers). Stronger baselines, or discussions of existing baselines at least, would greatly support their experiments.
- **Correlational studies (begging the question)**: My biggest confusion on LCS is how the paper grounds the notion of ICL effectiveness. It is unclear what the authors are definied as ICL effectiveness. At some points, ICL effectiveness is grounded by $\Delta$, the performance change before and after ICL (e.g., Figure 1/2, Table 1, and the main argument of Section 3.2.1). However, Section 3.2.2 claims that performance change cannot reflect ICL effectiveness, which is measured by performance change just on a larger population. I find this reasoning circular and puts LCS on a weak foundation. For example, I have a hard time parsing this sentence in L414-416 without some implicit circular reasoning: "However, LCS does not generally increase or decrease with the number of shots but rather exhibits some degree of fluctuation. This is because the value of LCS is related to the inherent ICL effectiveness on a given model and dataset, while increasing shot number cannot affect the ICL effectiveness."

**Questions:**

1. In L49, the paper claims ICL performance-based metrics are not reliable. Can you provide practical examples where this is the case? Or cases where one has to perform ICL but obtaining labeled data is difficult?
2. Theorem 2 proof error: L1062-1066 incorrectly uses the law of total probability or am I mistaken? Each equation should have a $\hat{p}(X|Q)$ multiplied. The conclusion would still hold.
3. Can you report $R^2$ values for Figure 10? It is hard to tell that contextual relevance is a better predictor than other metrics. The difference is small that I would like to see this plot with other datasets (MATH or GSM8K). Without convincing evidence, it is hard to see LCS as a useful predictive tool.
4. “Bad cases” analysis (Fig. 3): while the stated conclusion "Even on data where ICL does not improve performance, LCS still reveals the ICL effectiveness"(L309) is true, this highlights one issue with LCS. LCS only provides a metric for (model, task). It is not predictive of ICL effectiveness of each demo (model, task, ICL demo), so even when LCS=1.06>threshold=0.2, there exist cases where ICL performance improvement $\Delta$ is equal to zero.
5. Table 2: This is a great result, but I would like to see the performances of these methods on the other tasks (GSM8K, ARC-C, etc.) to see how much LCS-based selection can ameliorate negative ICL effects. Furthermore, is IDS and GTR the state-of-the-art (training-free) ICL example selection method? I recall many works on this, and would like for the authors to compare against more diverse baselines, e.g., https://arxiv.org/pdf/2302.11042 or https://aclanthology.org/2023.emnlp-main.331.pdf.
6. L427 "multi-round iterative process is used to ensure the diversity and quality of the synthesized": can you describe in detail what this process is? Is it manual or is there some heuristics for diversity?

Misc.
1. Regression slope inconsistency: Which slope is LCS? Theorem 1 claims
$$
  I(X \to D\mid Q)=\frac{p(D\mid Q)}{p(X\mid Q)} I(D \to X\mid Q).
$$
Thus, if you regress $t = I(X \to D\mid Q)$ on $s = I(D \to X\mid Q)$, the slope is $\frac{p(D\mid Q)}{p(X\mid Q)}$. However, Eq. (3) defines $s_i:=I(d_i \to x_i\mid q_i)$ and $t_i:=I(x_i \to d_i\mid q_i)$ but then sets the slope as
$$
  r_{\hat p}=\frac{\sum (t_i-\bar t)(s_i-\bar s)}{\sum (t_i-\bar t)^2},
$$
which is the slope of $s$ on $t$, if I am not mistaken. Meanwhile, plots in Figs. 7–9 show fitted lines with the slope of $t$ on $s$.
2. In Section 2.2 the paper suddenly calls LCS the “Learning‑to‑Relevance Ratio”.
3. Appendix C.1 proof introduces a mysterious variable $K$.
4. L352: "does" -> "do". L42: "cross" -> "across"

---

> ### Author Response · Authors · 2025-11-16
> **Official Comment by Authors (1/3)**
>
> Thank you for recognizing our work! **We have revised the paper in light of your review, with all changes highlighted in blue**. Below are our responses to your questions:
>
> 1. *Unless we make a very strong “order-invariance” assumption, Eq. (2) does not apply to the standard AR language models used in the rest of the paper. (**Appendix E.4 in Revision**)*
> * We add a supplementary discussion of this point in Appendix E.4 of the revision. In this paper, we assume the exchangeability of order in our derivation because:
>     - Previous research suggests that this is a limitation of the models themselves; an ideal model should not be affected by order [1,2].
>     - Many studies have found that the influence of text order on ICL is diminishing [3]. There are also many influential works on ICL analysis that are based on this same assumption [4–6].
>     - Our experimental results are consistent with our theoretical conclusions, which support the reasonableness of this assumption.
> * Therefore, in this paper, we assume that the order of demonstrations and queries in the input does not affect performance. **This allows us to focus more on the fundamental nature of ICL effectiveness and how to measure it**. Our experimental results also validate our conclusions under this assumption.
>
> 2. *Token probabilities are multiplied to obtain a joint score, which is not a proper probabilistic transformation and breaks the direct connection to the mathematical analysis. (**Appendix E.4 in Reviaion**)*
> * **This is a widely used way of modeling probabilities**: almost all probability-based language-model papers model the probability of generating an answer as the product of the probabilities of each token in the answer [7-9]. Therefore, in this paper, we follow prior work and adopt this probability computation method.
>
> 3. *Theorem 2 assumes there exists a true demonstration that bears most of the weight in the conclusion unless the authors can empirically show that this assumption is not too strong. (**Section 2.3 in Revision**)*
> * This assumption does not require the demonstration to be optimal; it only requires that the real (human) demonstration is better than synthetic demonstrations. We have paraphrased the statement of Theorem 2: **if demonstration A is better than demonstration B, then the LCS computed using B will be lower than the LCS computed using A**, which is a weaker assumption. We have updated this formulation in the revision.
>
> 4. *The motivation for the loss is logarithmic (Eq. (2)), but the core quantity is later defined as a difference of probabilities, and the slope result linearly relates to the $I_p$ term. I do not quite understand what Section 2.1 is trying to show. (**Appendix E.5 in Revision**)*
> * As we state in Lines 79-82, **the main motivation of this paper is to measure the effectiveness of demonstrations by quantifying the loss reduction brought about by adding demonstrations**. Therefore, Section 2.1 primarily provides the definition of the loss.
> * We define $I_p(X \rightarrow D \mid Q)$ in terms of probabilities rather than logarithms to facilitate the subsequent derivation because the probability form makes it easier to relate different factors. In Appendix E.5, we further explain that $I_p(X \rightarrow D \mid Q)$ is positively correlated with the degree of loss reduction, i.e., $I_p(X \rightarrow D \mid Q)$ defined in this way can also reflect the extent of loss decrease, which is consistent with our motivation for using loss reduction to measure ICL effectiveness. The consistency between our experimental results and the theoretical predictions further confirms the soundness of this assumption.
>
> 5. *Stronger baselines, or at least a more thorough discussion of existing baselines, would substantially strengthen the empirical case. (**Table 2 in Revision**)*
>
> | Method      | GSM8K | MATH | ARC-C | MMLU-Pro | FinQA | Amazon |
> |------------|:-----:|:----:|:-----:|:--------:|:-----:|:------:|
> | Zero-Shot  | 86.4  | 48.4 | 82.1  | 50.4     | 49.7  | 63.5   |
> | BM25       | 84.2  | 50.8 | 80.2  | 53.0     | 54.6  | 68.5   |
> | GTR        | 82.1  | 50.8 | 80.5  | 53.5     | 55.0  | 68.5   |
> | Yang et al | 84.2  | 50.2 | 80.9  | 53.3     | 55.0  | 69.0   |
> | Influence  | 83.9  | 51.0 | 81.3  | 52.4     | 54.6  | 69.5   |
> | IDS        | 85.3  | 50.4 | 82.4  | 52.4     | 54.6  | 68.0   |
> | Ours       | **86.4** | **51.2** | **82.5** | **54.3** | **55.1** | **70.0** |
>
> * In the revision, we add GSM8K and ARC-Challenge results to Table 2 and include the two baselines you suggested.

---

> > ### Author Response · Authors · 2025-11-16
> > **Official Comment by Authors (2/3)**
> >
> > 6. *My biggest confusion about LCS is how the paper defines ICL effectiveness. Sometimes they use $\Delta$ to characterize it, but then claim that performance change cannot capture ICL effectiveness, while ICL effectiveness is again measured in terms of a larger overall performance change. I feel this is logically circular, which weakens the foundation of LCS. For example, I find it hard to read L414–416. (**Lines 242-246 in Revision**)*
> > * The point you raise is indeed very important. In Section 3.2, we also use performance change ($\Delta$) as a measure of ICL effectiveness because we want to show that LCS can reflect ICL effectiveness not only on datasets where $\Delta$ accurately reflects ICL effectiveness (corresponding to the results in Table 1), but also on datasets where $\Delta$ fails to accurately capture ICL effectiveness (corresponding to the results in Figure 3). In this way, **we demonstrate that LCS is a better metric of ICL effectiveness than performance change**.
> > * Regarding the sentence you quoted from L414–416, it can be interpreted as follows: given a fixed dataset and model, **ICL effectiveness is an inherent capability of the model, and this capability does not change with the number of shots. Different metrics, however, vary in how reliably they reflect this ability**. Performance changes yield different values at different numbers of shots, making it difficult to reliably determine whether ICL is effective for a given model–dataset pair. Our proposed metric, in contrast, remains stable as the number of shots changes, indicating that it can reliably reflect ICL effectiveness.
> >
> > 7. *L49 states that “metrics based on ICL performance are unreliable.” Could you provide some real-world scenarios as examples? Or provide a scenario where ICL must be used, but labeled data is hard to obtain?*
> > * We provide an example in Figure 3. Specifically, we select bad cases of Llama 3.1-8B with ICL on GSM8K and Amazon Review. In these cases, the performance gain from ICL is $0$, indicating that “metrics based on ICL performance are unreliable,” whereas LCS can still reflect ICL effectiveness in these bad cases, demonstrating the reliability of LCS.
> > * Here is an example where ICL must be used, but labeled data are hard to obtain. In real power-grid applications, due to limited computational resources, it may be impossible to train a model from scratch; thus, ICL is used to help the model adapt to the power-grid domain. However, power-grid data is often sensitive or private, making it difficult to obtain labeled data.
> >
> > 8. *Lines 1062–1066 seem to misuse the law of total probability. (**Appendix C.4 in Revision**)*
> > * Thank you for pointing this out! We have corrected this in the revision.
> >
> > 9. *It is currently hard to see that contextual relevance truly outperforms other metrics in Figure 10. (**Lines 147-149 and Appendix F.2 in Revision**)*
> > * As discussed in Lines 147-149, **the purpose of Figure 10 is not to prove that contextual relevance is superior to other metrics, but to show that our definition of Contextual Relevance can faithfully reflect the relevance between demonstrations and user queries**, thereby aligning with the statement in Lines 145-146, where we define ICL effectiveness as the ratio between loss reduction and the relevance between demonstrations and the query. In Figure 10, the trend of contextual relevance is consistent with common relevance metrics such as BM25 and cosine similarity, showing that our probability-based $I_p(D \rightarrow X \mid Q)$ can also reflect relevance.
> > * You can see the numerical values of the fitted regression line above the points in each subfigure of Figure 10. In the revision, we also add analogous Figure 10 plots for MATH and GSM8K.
> >
> > 10. *LCS only provides a metric at the (model, task) level; it cannot predict the ICL outcome for each specific tuple of (model, task, ICL demonstrations).*
> > * Yes, as we discuss in Lines 79-82, **the goal of this work is not to measure whether each individual demonstration is effective, but rather, given a fixed model and task, to analyze the factors that influence ICL effectiveness and to guide users in deciding whether to annotate demonstrations**. The problem you mention is that predicting the effect of a single demonstration is closer to work on demonstration selection, which is different from the objective of our paper.

---

> > > ### Author Response · Authors · 2025-11-16
> > > **Official Comment by Authors (3/3)**
> > >
> > > 11. *I would like to see how these methods perform on other tasks to assess how far LCS-based selection can mitigate negative ICL effects. Also, are IDS and GTR really SOTA training-free ICL demonstration selection methods? There is a lot of work in this area; I hope the authors can compare it with a broader set of baselines. (**Table 2 in Revision**)*
> > >
> > > | Method      | GSM8K | MATH | ARC-C | MMLU-Pro | FinQA | Amazon |
> > > |------------|:-----:|:----:|:-----:|:--------:|:-----:|:------:|
> > > | Zero-Shot  | 86.4  | 48.4 | 82.1  | 50.4     | 49.7  | 63.5   |
> > > | BM25       | 84.2  | 50.8 | 80.2  | 53.0     | 54.6  | 68.5   |
> > > | GTR        | 82.1  | 50.8 | 80.5  | 53.5     | 55.0  | 68.5   |
> > > | Yang et al | 84.2  | 50.2 | 80.9  | 53.3     | 55.0  | 69.0   |
> > > | Influence  | 83.9  | 51.0 | 81.3  | 52.4     | 54.6  | 69.5   |
> > > | IDS        | 85.3  | 50.4 | 82.4  | 52.4     | 54.6  | 68.0   |
> > > | Ours       | **86.4** | **51.2** | **82.5** | **54.3** | **55.1** | **70.0** |
> > >
> > > * In the revision, we add GSM8K and ARC-Challenge results to Table 2 and include the two baselines you suggested. IDS and GTR are not the current SOTA because **the main goal of this paper is not to propose a new demonstration selection method, but to analyze the ICL effectiveness**. Section 3.2.3 is intended to validate the usefulness of LCS, and the performance gains we observe indeed confirm the effectiveness of LCS.
> > >
> > > 12. Could you describe the synthesis process in more detail? Is it manual, or does it follow some heuristic selection strategy? (**Lines 445-446 in Revision**)*
> > > * Concretely, the synthesis is a multi-iteration iterative process: in each iteration, we provide the model with the task definition and the synthetic results from the previous iteration (empty in the first iteration), ask the model to generate demonstrations, and set the sampling size to 8 to ensure diversity. **We also provide the synthesis prompt in Appendix D.3**.
> > >
> > > 13. *Inconsistency in the regression slope. (**Equation 3 in Revision**)*
> > > * Thank you for pointing out this typo! We have fixed it in the revision by swapping the roles of $s_i$ and $t_i$. Indeed, what we should be computing is the slope of $I(x \to d \mid q)$ with respect to $I(d \to x \mid q)$.
> > >
> > > 14. *In Section 2.2, the paper suddenly refers to LCS as “Learning-to-Relevance Ratio.” (**Line 171 in Revision**)*
> > > * Thank you for catching this typo! It should be “Learning-to-Context Slope.”
> > >
> > > 15. *A mysterious variable $K$ appears in the proof in Appendix C.1. (**Appendix C.1 in Revision**)*
> > > * This $K$ should not appear; we have removed it in the revision.
> > >
> > > [1] Rethinking Invariance in In-Context Learning, ICLR 2025.
> > >
> > > [2] Addressing Order Sensitivity of In-Context Demonstration Examples in Causal Language Models, ACL 2024.
> > >
> > > [3] Rapid Selection and Ordering of In-Context Demonstrations via Prompt Embedding Clustering, ICLR 2025.
> > >
> > > [4] What and How does In-Context Learning Learn? Bayesian Model Averaging, Parameterization, and Generalization, ICLR 2024.
> > >
> > > [5] Can Generative AI Solve Your In-Context Learning Problem? A Martingale Perspective, ICLR 2025.
> > >
> > > [6] From Few to Many: Self-Improving Many-Shot Reasoners through Iterative Optimization and Generation, NeurIPS 2025.
> > >
> > > [7] Neural Probabilistic Language Model, 2003.
> > >
> > > [8] Sequence to Sequence Learning with Neural Networks, 2014.
> > >
> > > [9] Improving Language Understanding by Generative Pre-Training, 2018.

---

> ### Author Response · Authors · 2025-11-28
>
> Dear Reviewer, we have revised the paper based on your review and submitted a response, and we sincerely hope that our changes address your concerns. As there are now fewer than five days remaining before the rebuttal period ends, we would be truly grateful for your feedback and sincerely look forward to your reply.

---

### Official Review · Reviewer_EHcL · 2025-11-05

**Soundness:** 3
**Presentation:** 3
**Contribution:** 2
**Rating:** 6
**Confidence:** 4

**Summary:**

The paper proposes Learning to Context Slope, LCS, as a metric to evaluate the effectiveness of in context learning. The idea is to measure how much the loss decreases when we add demonstrations, as a function of how relevant those demonstrations are to the input. Formally, the authors define two quantities per triple Q as the input question, X as the output label, and D as the demonstrations. They define two Bayesian relationship terms:

Learning gain $t = I(X \rightarrow D \mid Q)$

Contextual relevance $s = I(D \rightarrow X \mid Q).$

Over many triples, they fit a line from t to s using least squares, and the slope is the definition of LCS. A larger slope means that increases in relevance translate into larger loss reductions. The paper argues that LCS is more reliable than simple performance deltas because it uses continuous loss even when the final answer is wrong. It also claims better attribution. Low LCS can arise from weak contextual alignment, meaning the model does not extract the right signals from demonstrations, or from strong output calibration, meaning the model already verifies answers without demonstrations.

**Strengths:**

- Clear motivation. Performance based evaluation can be noisy and hard to attribute. LCS targets the underlying loss dynamics.
- Simple mathematical core. The link between learning gain and contextual relevance is expressed as a slope that practitioners can estimate with standard scoring. Because it uses token level loss, LCS can detect progress that accuracy metrics miss.
- Although interactions within D cannot be measured precisely and a microscopic view is not provided as noted in weakness one, when order and interactions between demonstration samples are treated as fixed, LCS appears to provide a useful global view.

**Weaknesses:**

- LCS is a set level first order summary. It does not model higher order interactions between multiple demonstrations, such as redundancy, synergy, order, and position effects. The paper treats k shot by splitting into k points, which ignores interactions.
- Assumptions in theory. Theorems rely on modeling choices and oracle versus empirical probabilities. They are valid under stated conditions, but the practical gap due to estimation error and prompt format choices remains.
- Computational cost. Estimating s and t requires multiple scoring passes with consistent templates and length normalization. This can be heavy for large models and many demonstrations.
- Interpretability of magnitude. The paper reports that LCS correlates with performance, but a universal threshold for large positive, medium, or small is not fully established in the main text.

**Questions:**

- Naming inconsistency. (minor error I guess) The text uses Learning to Relevance Ratio (LCS) at line 162 while the main name is Learn to Context Slope (LCS).
- When the base model already has strong output calibration, I guess LCS can be low and ICL may not help. The paper frames this as expected, but does this limit usefulness in easy tasks or with very strong models.
- ICL can also learn well for adversarial or label flipped scenarios. If demonstrations are systematically misleading, I presume they can learn misbehavior quite well as well. It is unclear how stable LCS remains in practice and whether it can output robust and reliable conclusions. Robustness under adversarial or flipped labels. If demonstrations encode a wrong mapping that the model can learn, how do s, t, and LCS behave relative to accuracy and loss on the true label. Can the authors provide an experiment that stresses LCS in such cases.
- Thresholds and portability. The paper mentions actionable thresholds. Can the authors provide numeric guidance that transfers across datasets and models, for example a range of LCS values that usually indicate effective ICL.

---

> ### Author Response · Authors · 2025-11-16
> **Official Comment by Authors (1/2)**
>
> Thank you for your positive assessment of our work! **We have revised the paper in response to your review, and all changes are highlighted in blue**. Below, we address your specific questions point by point.
>
> 1. *LCS does not model higher order interactions between multiple demonstrations. (**Appendix E.4 in Revision**)*
> * We add a supplementary discussion of this point in Appendix E.4 of the revision. In this paper, we assume the exchangeability of order in our derivation because:
>     * Previous research suggests that this is a limitation of the models themselves; an ideal model should not be affected by order [3, 4].
>     * Many studies have found that the influence of text order on ICL is diminishing [5]. There are also many influential works on ICL analysis that are based on this assumption [6–8].
>     * Our experimental results are consistent with our theoretical conclusions, which support the reasonableness of this assumption.
> * Therefore, in this paper, we assume that the order of demonstrations and queries in the input does not affect performance. **This allows us to focus more on the fundamental nature of ICL effectiveness and how to measure it**. Our experimental results also validate our conclusions under the above assumption.
>
> 2. *The practical gap due to estimation errors and prompt format choices remains. (**Appendix C.3**)*
> * In Appendix C.3, **we analyze how such errors affect the computed results**, and based on this analysis, we show that our theoretically motivated analysis of computing LCS can, in principle, reduce the influence of these errors.
> * Indeed, in practical computation, estimation errors, prompt-format choices, and other factors will inevitably introduce errors. However, since the experimental results in Section 3.2 are consistent with the theoretical predictions, we believe that this level of error is acceptable to some extent. Completely eliminating such errors, however, will require further work in future research.
>
> 3. *Estimating $s$ and $t$ requires heavy computation for large models and numerous demonstrations. (**Appendix E.3**)*
> * We discuss the computational efficiency of LCS in Appendix E.3. Although our time cost is higher than that of a single inference, **our primary motivation is to propose measuring ICL effectiveness and the factors affecting it, based on which we can guide users to determine whether to label demonstrations rather than to perform inference as efficiently as possible**. Therefore, we consider the additional time overhead to be acceptable.
>
> 4. *The paper reports that LCS correlates with performance, but a universal threshold for large positive, medium, or small is not fully established in the main text. (**Appendix E.2**)*
> * In Appendix E.2, we provide an empirical threshold: **the relationship between LCS and 0.2 can be used to judge the degree of ICL effectiveness**. However, given the varying needs in practical applications, it is preferable for users to determine for themselves what magnitude of LCS is acceptable for their particular use case.
>
> 5. *The text uses the Learning to Relevance Ratio (LCS) at line 162. (**Line 171 in Revision**)*
> * Thank you for pointing this out! We have fixed this issue in the revision.
>
> 6. *When the base model already has strong output calibration, I suppose LCS can be low, and ICL may not help. The paper frames this as expected, but does this limit its usefulness in easy tasks or with very strong models.*
> * **The fact that LCS is low on simple tasks or for very strong models actually supports the effectiveness of LCS**: it indicates that LCS accurately reflects the low effectiveness of ICL in these scenarios. In such situations, the condition in Theorem 1 corresponds to a large $p(X \mid Q)$, meaning that the model can already solve the given dataset well, and LCS should indeed be low. Therefore, the practicality of LCS is not restricted in these cases.

---

> > ### Author Response · Authors · 2025-11-16
> > **Official Comment by Authors (2/2)**
> >
> > 7. *If demonstrations encode a wrong mapping that the model can learn, how do s, t, and LCS behave relative to accuracy and loss on the true label? (**Appendix F.5 in Revision**)*
> > * In Appendix F.5 of the revision, we have added experiments on label flipping. The results are shown below:
> >
> > | Dataset   | Type     | Llama2-7b $\Delta$ | Llama2-7b LCS | Llama3.1-8b $\Delta$ | Llama3.1-8b LCS | Llama-R1-8b $\Delta$ | Llama-R1-8b LCS |
> > |----------|----------|-------------|-----------------------|---------------|-------------------------|---------------|-------------------------|
> > | MATH     | Real     | +9.6        | 1.03                  | +2.4          | 0.34                    | -1.2          | 0.09                    |
> > | MATH     | Mismatch | +4.2        | 0.72                  | +2.0          | 0.22                    | -1.4          | 0.03                    |
> > | MMLU-Pro | Real     | +5.5        | 0.64                  | +2.6          | 0.52                    | -5.5          | -0.04                   |
> > | MMLU-Pro | Mismatch | +4.3        | 0.52                  | +1.9          | 0.30                    | -5.0          | -0.12                   |
> >
> > * Specifically, we observe that after flipping the labels, the model can still improve its performance via ICL, which is consistent with conclusions from prior work, and LCS accurately reflects this phenomenon. However, compared with the case of true labels, the performance of LCS on flipped labels is relatively lower, which is in line with the conclusions of our Theorem 2: the quality of demonstrations with flipped labels is lower than that of demonstrations with true labels, leading to a correspondingly lower LCS.
> >
> > 8. *Can the authors provide numeric guidance that transfers across datasets and models.*
> > * Please refer to the answer for Q4.
> >
> > [1] Rethinking Invariance in In-Context Learning, ICLR 2025.
> >
> > [2] Addressing Order Sensitivity of In-Context Demonstration Examples in Causal Language Models, ACL 2024.
> >
> > [3] Rapid Selection and Ordering of In-Context Demonstrations via Prompt Embedding Clustering, ICLR 2025.
> >
> > [4] What and How does In-Context Learning Learn? Bayesian Model Averaging, Parameterization, and Generalization, ICLR 2024.
> >
> > [5] Can Generative AI Solve Your In-Context Learning Problem? A Martingale Perspective, ICLR 2025.
> >
> > [6] From Few to Many: Self-Improving Many-Shot Reasoners through Iterative Optimization and Generation, NeurIPS 2025.

---

> ### Author Response · Authors · 2025-11-28
>
> Dear Reviewer, we have revised the paper based on your review and submitted a response, and we sincerely hope that our changes address your concerns. As there are now fewer than five days remaining before the rebuttal period ends, we would be truly grateful for your feedback and sincerely look forward to your reply.

---

### Official Review · Reviewer_VfMS · 2025-11-05

**Soundness:** 3
**Presentation:** 3
**Contribution:** 3
**Rating:** 6
**Confidence:** 3

**Summary:**

This work proposes a new metric called Learning-to-Context Slope (LCS) to measure how effectively large language models (LLMs) perform in-context learning (ICL). Compared to traditional evaluations that simply measure task performance, LCS quantifies the relationship between learning gain (loss reduction from demonstrations) and contextual relevance (alignment between demonstrations and input). This continuous, loss-based metric captures ICL effectiveness even when outputs are incorrect, provides clearer attribution to model limitations such as weak contextual alignment or excessive output calibration, and works without labeled data through synthetic evaluation. Experiments across eight datasets and multiple LLMs show that LCS correlates strongly with performance gains and reveals that ICL effectiveness depends mainly on a model’s ability to align contextually with demonstrations rather than raw performance.

**Strengths:**

The proposed LCS metric is a novel and potentially useful idea. As this idea goes beyond performance-based evaluations and offers continuous, loss-based measure that remains informative even when model outputs are incorrect, the metric allows for a more refined analysis of ICL capabilities. Moreover, LCS works without labeled data through synthetic evaluation, making it applicable when data is limited. The experiments across multiple datasets and models validate its robust correlation with actual performance improvements, demonstrating both utility.

As far as I know, the idea and the metric presented in this work is novel, and I feel it is potentially very valuable for better quantifying the progress of ICL methodology.

**Weaknesses:**

LCS primarily measures correlation, not causation as a high slope indicates association between loss reduction and context relevance, but does not directly prove that demonstrations cause better learning. Also, the theory and the metric's interpretability depends on strong modelling assumptions.

Also, the experiments could be expanded to a broader range of model families.

**Questions:**

Can the authors clarify the intuition behind modeling ICL effectiveness as a slope? Why is a linear relationship assumed between loss decrease and contextual relevance?

---

> ### Author Response · Authors · 2025-11-16
>
> Thank you for your positive evaluation of our work! **We have revised the manuscript in light of your review, with all changes highlighted in blue**. Below are our responses to your questions:
>
> 1. *LCS primarily measures correlation, which does not directly prove that demonstrations cause better learning. Also, the theory and the metric's interpretability depend on strong modeling assumptions. (**Appendix E.4 in Revision**)*
> * As we discuss in Lines 43-44, **the goal of this work is not to measure whether each individual demonstration is effective, but rather, given a fixed model and task, to analyze the factors that influence ICL effectiveness and to guide users in deciding whether to annotate demonstrations**. The problem you mention is that predicting the effect of a single demonstration is closer to work on demonstration selection, which is different from the objective of our paper.
> * You can refer to the answer to Q3, which indicates that **some of our modeling assumptions are not based on personal intuition but on theoretical analysis**. Besides, in Appendix E.4, we further justify the reasonableness of our other modeling assumptions. Specifically, the assumptions we use are widely adopted in related works [1,2], and the experimental results are also consistent with our theoretical expectations, which supports the validity of these assumptions.
>
> 2. *The experiments could be expanded to a broader range of model families. (**Appendix F.3 in Revision**)*
> * We adapt experiments beyond the models of the main experiment in Appendix F.3. **In the revision, we also added experiments on Qwen3-8b and Ministral-8B-Instruct-2410 (Ministral-8b), covering different model families**. All of these experimental results are shown below. The results are consistent with our existing conclusions and further substantiate them.
>
> | **Model**       | **MATH**            |                    | **FinQA**           |                    | **Amazon**          |                    |
> |-----------------|---------------------|--------------------|---------------------|--------------------|---------------------|--------------------|
> |                 | $\Delta$            | $r_{\hat{p}}x + b$ | $\Delta$            | $r_{\hat{p}}x + b$ | $\Delta$            | $r_{\hat{p}}x + b$ |
> | Llama3.1-8b     | $+2.4$              | $0.34 x - 0.00$    | $+4.9$              | $0.82 x - 0.06$    | $+5.0$              | $0.94 x - 0.09$    |
> | Llama3.1-70b    | $-3.6$              | $-0.13 x - 0.04$   | $+7.6$              | $0.77 x - 0.07$    | $+16.0$             | $0.79 x - 0.18$    |
> | Qwen2.5-7b      | $+2.2$              | $0.81x-0.16$       | $+4.9$              | $0.29x - 0.09$     | $+27.0$             | $0.42x+0.00$       |
> | Qwen3-8b        | $-1.4$              | $-0.21x-0.08$      | $+6.7$              | $0.53x+0.21$       | $-1.0$              | $0.03x-0.13$       |
> | Ministral-8b    | $-1.8$              | $0.04x-0.02$       | $+4.8$              | $0.71x-0.04$       | \textbf{$+4.0$}     | $0.69x-0.11$       |
>
> 3. *Can the authors clarify the intuition behind modeling ICL effectiveness as a slope? Why is a linear relationship assumed between loss decrease and contextual relevance?*
> * Our modeling of the relationship between loss decrease and context relevance as linear, and of LCS as a slope, is **not based on an assumption but on the theoretical derivation in Theorem 1**, in which we prove that loss decrease and context relevance satisfy a linear relationship.
> * As discussed in Lines 79–81, we use loss decrease and context relevance to measure the effectiveness of ICL because, to achieve the same loss decrease on a given dataset, a model with less effective ICL requires examples with higher relevance. The experimental results in Section 3 also show that, compared with changes in task performance, LCS can better evaluate the effectiveness of ICL.
>
> [1] What and How does In-Context Learning Learn? Bayesian Model Averaging, Parameterization, and Generalization, ICLR 2024.
>
> [2] Can Generative AI Solve Your In-Context Learning Problem? A Martingale Perspective, ICLR 2025.

---

> ### Author Response · Authors · 2025-11-28
>
> Dear Reviewer, we have revised the paper based on your review and submitted a response, and we sincerely hope that our changes address your concerns. As there are now fewer than five days remaining before the rebuttal period ends, we would be truly grateful for your feedback and sincerely look forward to your reply.

---

### Author Response · Authors · 2025-11-28
**Rebuttal Summary for AC**

Dear AC,

Thank you very much for your contribution to the conference! To help reduce your workload, we briefly summarize our paper, the reviewers’ core concerns, and our responses.

---

This paper studies when and why in-context learning (ICL) is effective and proposes **Learning-to-Context Slope (LCS)**. Our main conclusions are:
1. We model ICL effectiveness as the slope between learning gain (loss decrease after adding demonstrations) and contextual relevance, giving a continuous LCS metric that can evaluate ICL via token-level loss even when final answers are wrong.
2. We theoretically show a linear relation between learning gain and contextual relevance and decompose LCS into contextual alignment and output calibration, explaining why ICL succeeds or fails.

---

Strengths highlighted by reviewers:
1. Novel continuous ICL-effectiveness metric. (VfMS, bnZL)
2. Works with unlabeled and synthetic data. (VfMS)
3. Clear mathematical formulation and intuition. (EHcL, yHFq)
4. Global loss-based view beyond accuracy. (EHcL, bnZL)
5. Broad experiments and strong performance correlation. (VfMS, bnZL, yHFq)
6. Practical guidance for demo annotation and selection. (VfMS, bnZL)

---

Main concerns and our responses:

1. “LCS only reflects correlation and the definition of ICL effectiveness may be circular.” (VfMS, bnZL, yHFq)
* Our goal is **overall ICL diagnosis and attribution** for a fixed model and task, not per-demo causal effects. Performance gain is used only as a reference when reliable; on cases where performance gain is uninformative, LCS still captures ICL behavior, showing it is more stable than accuracy-based metrics.

2. Strength of theoretical assumptions (exchangeability, autoregressive implementation, log-loss vs probability form). (VfMS, EHcL, bnZL)
* We explicitly discuss the exchangeability assumption and cite prior ICL theory using the same setting. We also prove that our probability-based definition is monotonically related to loss reduction, and the empirical results match the theory.

3. Assumptions in theorems, pseudo-probabilities, and theory–practice gap. (EHcL, bnZL)
* Multiplying token probabilities is standard in language modeling; we analyze in the appendix how length normalization and estimation error affect LCS and show our procedure mitigates them. We also weaken Theorem 2 to a realistic monotonic assumption: higher-quality demonstrations yield higher LCS.

4. Limited experiments and demo-selection baselines. (VfMS, EHcL, bnZL)
* We add experiments on Qwen3-8B, Ministral-8B and **label-flipping** scenarios, showing LCS distinguishes demonstration quality when the model learns correctly or incorrectly. For demo selection we add tasks (GSM8K, ARC-C) and stronger baselines (Yang et al., Influence). Our method is overall comparable to or better than them; this section is meant to validate LCS, not propose a new SOTA selector.

5. Interpretability of LCS values and behavior on simple tasks or strong models. (EHcL, bnZL)
* We provide an empirical threshold around **0.2**, to be adapted per application. On simple tasks or well-calibrated models the marginal benefit of demonstrations is small, so a low LCS correctly indicates that extra ICL annotation is unnecessary.

6. Computational cost, black-box models, and need for synthetic-data experiments. (EHcL, yHFq)
* LCS is designed for **offline diagnosis and decision making**, not per-query inference, so extra computation is acceptable. For black-box models we approximate log-likelihoods and compute LCS from the fitted values; results still match ICL effectiveness. Synthetic experiments show that in label-free settings LCS from synthetic demonstrations follows the same trends as with real data and can guide whether human annotation is worthwhile.

7. Organization, and writing details. (yHFq, bnZL)
* We refine the organization, and fix all notation and typographical issues raised.

Thank you again for your efforts! We hope this summary helps reduce your workload.

Best regards

Authors

---

### Note · Program_Chairs · 2026-01-17
**Submission Desk Rejected by Program Chairs**

The following references in this submission do not refer to real documents and/or have major errors in bibliographic information:

 Omer Levy, Gabriel Poesia, Sewon Min, Romain Paulus, Luke Zettlemoyer, and Mike Lewis. Diverse demonstrations improve in-context compositional generalization. arXiv preprint arXiv:2211.12703, 2022.